# Macrophage VLDLR mediates obesity-induced insulin resistance with adipose tissue inflammation

Kyung Cheul Shin[1], Injae Hwang[1], Sung Sik Choe[1], Jeu Park[1], Yul Ji[1], Jong In Kim[1], Gha Young Lee[2], Sung Hee Choi[2,3], Jianhong Ching[4], Jean-Paul Kovalik[4] & Jae Bum Kim[1]

Obesity is closely associated with increased adipose tissue macrophages (ATMs), which contribute to systemic insulin resistance and altered lipid metabolism by creating a pro-inflammatory environment. Very low-density lipoprotein receptor (VLDLR) is involved in lipoprotein uptake and storage. However, whether lipid uptake via VLDLR in macrophages affects obesity-induced inflammatory responses and insulin resistance is not well understood. Here we show that elevated VLDLR expression in ATMs promotes adipose tissue inflammation and glucose intolerance in obese mice. In macrophages, VLDL treatment upregulates intracellular levels of C16:0 ceramides in a VLDLR-dependent manner, which potentiates pro-inflammatory responses and promotes M1-like macrophage polarization. Adoptive transfer of VLDLR knockout bone marrow to wild-type mice relieves adipose tissue inflammation and improves insulin resistance in diet-induced obese mice. These findings suggest that increased VLDL-VLDLR signaling in ATMs aggravates adipose tissue inflammation and insulin resistance in obesity.

[1] Department of Biological Sciences, National Creative Research Initiatives Center for Adipose Tissue Remodeling, Institute of Molecular Biology and Genetics, Seoul National University, Seoul 08826, Korea. [2] Department of Internal Medicine, Seoul National University College of Medicine and Seoul National University Bundang Hospital, Seongnam 13620, Korea. [3] Department of Internal Medicine, Seoul National University College of Medicine, Seoul 03080, Korea. [4] Cardiovascular and Metabolic Disorders, Duke-NUS Medical School, Singapore 169857, Singapore. Correspondence and requests for materials should be addressed to J.B.K. (email: jaebkim@snu.ac.kr)

Obesity is characterized by chronic and low-grade inflammation accompanied with macrophage accumulation in adipose tissue, eventually leading to metabolic disorders including insulin resistance and type 2 diabetes[1, 2]. Adipose tissue macrophages (ATMs) are key players in adipose tissue inflammatory responses in obesity[3–5]. In lean animals, the large number of ATMs is composed of alternatively activated (M2-like) macrophages expressing high levels of interleukin (IL)-4, -10, -13, and arginase (ARG) 1 that are associated with insulin sensitivity[6, 7]. Although it has been shown that M2-like macrophages might produce catecholamines to enhance adaptive thermogenesis[8–10], a very recent study reported that M2-like macrophages do not produce catecholamines[11]. These controversial findings are needed to be further investigated. In contrast, in obese animals, the population of classically activated (M1-like) macrophages is rapidly increased in adipose tissue[12, 13]. M1-like ATMs secrete numerous pro-inflammatory cytokines, such as tumor necrosis factor alpha (TNFα), and MCP-1, which aggravates adipose tissue inflammation and insulin resistance in obesity[14, 15]. In obese adipose tissue, pro-inflammatory cytokines secreted from M1-like ATMs induce adipokine dysregulation and impair insulin action to confer systemic insulin resistance[16, 17]. Thus, the imbalance between M1- and M2-like ATMs plays an important role to modulate pro-inflammatory responses in obese adipose tissue.

M1- or M2-like polarization of ATMs has been attributed to dynamic changes in adipose tissue microenvironments[18].

Concurrent with the expansion of adipose tissue in obesity, ATMs participate in adipose tissue remodeling by storing surplus lipid metabolites, giving rise to a subpopulation of lipid-laden ATMs[19–22]. Recent studies have shown that cytotoxic lipid species, such as free cholesterol, and short-chain saturated fatty acids, are elevated, whereas protective lipid metabolites, such as long-chain polyunsaturated fatty acids, are decreased in the lipid-laden ATMs of obese mice[21, 23]. Furthermore, lipid-overloaded macrophages in obese adipose tissue stimulate pro-inflammatory cytokines such as TNFα, steering to insulin resistance[24]. Also, it has been reported that macrophages are able to produce anti-inflammatory lipid metabolites such as DHA and EPA[25]. These findings suggest that alteration of lipid metabolism in ATMs would be crucial to induce inflammatory responses and insulin resistance in obesity.

In plasma, essential lipid metabolites, such as cholesterol and triglycerides, are circulated in the form of lipoproteins[26]. Major triglyceride-carrying lipoproteins are very low-density lipoprotein (VLDL) and chylomicron[27]. VLDL receptor (VLDLR) has a pivotal role to uptake VLDL and chylomicron through receptor-mediated endocytosis or lipoprotein lipase (LPL)-dependent lipolysis[28, 29]. VLDLR, a member of the low-density lipoprotein (LDL) receptor (LDLR) family, is abundantly expressed in adipose tissue, heart, kidneys, and skeletal muscle[28, 29]. Patients with VLDLR mutations exhibit low body mass index (BMI) compared to normal subjects[30, 31]. Similarly, VLDLR-deficient mice are

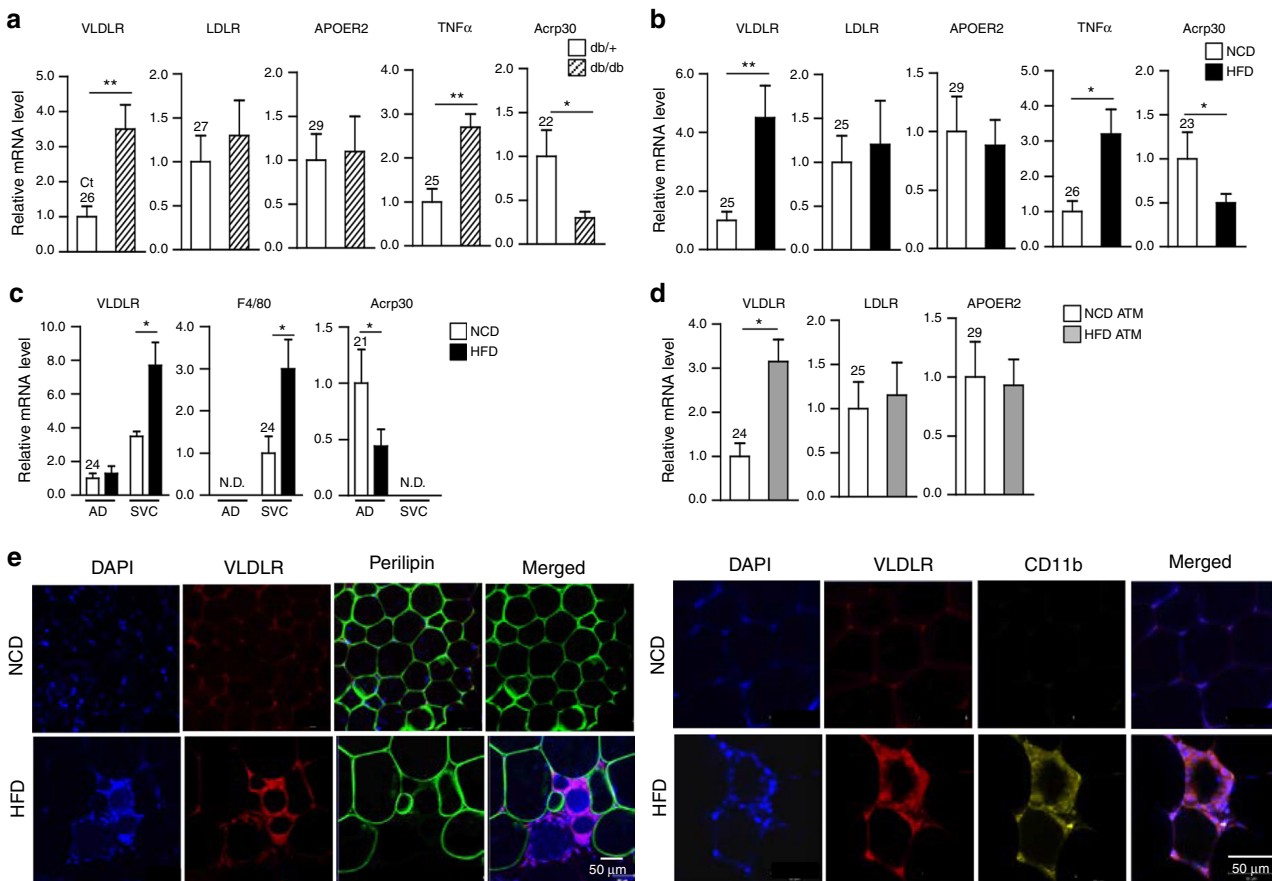

**Fig. 1** VLDLR expression is elevated in ATMs from obese mice. **a**, **b** Relative mRNA level of VLDLR in adipose tissue of **a** lean (*db/+*) (*n* = 4) and obese (*db/db*) (*n* = 5) mice and **b** normal chow diet (NCD)-fed (*n* = 6) and high-fat diet (HFD)-fed (*n* = 7) mice as measured by RT-qPCR. **c** Relative mRNA level of VLDLR in adipocyte (AD) and stromal vascular cell (SVC) fractions from NCD-fed (*n* = 5) and HFD-fed (*n* = 5) mice. **d** Relative mRNA level of VLDLR in sorted ATMs (F4/80⁺, CD11b⁺) from NCD-fed (*n* = 12) and HFD-fed (*n* = 9) mice. **a–d** Each mRNA level was normalized to cyclophilin mRNA. Data represent the mean ± SD. \**P* < 0.05, \*\**P* < 0.01, Student's *t*-test. **e** VLDLR protein was monitored in ATMs from NCD-fed and HFD-fed mice. Whole-mount immunohistochemistry analysis of the nucleus (blue), VLDLR (red), perillipin (green), and CD11b (yellow) in EATs from NCD-fed and HFD-fed mice

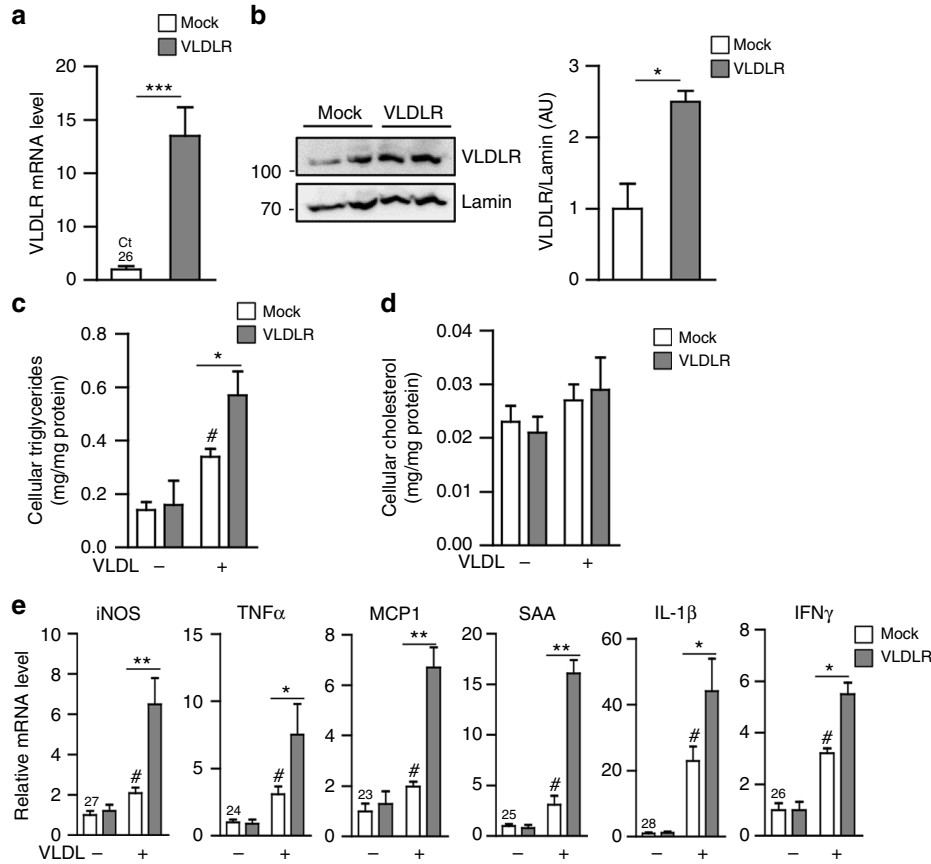

**Fig. 2** Macrophages overexpressing VLDLR accumulate triglycerides and induce expression of pro-inflammatory genes in the presence of VLDL. Peritoneal macrophages were transfected with pcDNA3.1-mock (Mock) vector or pcDNA3.1-VLDLR (VLDLR) expression vector. Total RNA was isolated to analyze VLDLR mRNA expression (**a**) and whole cell lysates were extracted for western blot analysis (**b**). **c**–**e** Human VLDL (30 μg/ml) was treated to cultures of peritoneal macrophages that had been transfected or not with VLDLR overexpression vector. Intracellular triglyceride (**c**) and cholesterol (**d**) levels in peritoneal macrophages. **e** mRNA levels of pro-inflammatory genes were analyzed with (+) or without (−) human VLDL challenges in peritoneal macrophages. Each mRNA level was normalized to cyclophilin mRNA. Data represent mean ± SD. #$P < 0.05$ vs. VLDL; *$P < 0.05$, **$P < 0.01$, ***$P < 0.001$, Student's $t$-test

protected from high fat diet (HFD)-induced obesity[32]. Furthermore, VLDLR-deficient mice exhibit improved glucose intolerance upon HFD, accompanied with alleviated inflammation and ER stress in adipose tissue[33]. In addition, it has been suggested that VLDL might influence cellular inflammatory responses in macrophages, thereby potentiating metabolic complications, such as atherosclerosis and diabetes[34, 35]. However, it remains largely unknown whether VLDLR-mediated VLDL uptake in macrophages is an important factor in mediating adipose tissue inflammation and insulin resistance in obesity.

In this study, we demonstrate that VLDLR is elevated in obese ATMs, and promotes adipose tissue inflammation by upregulating ceramide production and facilitating M1-like macrophage polarization. Moreover, bone marrow transplantation (BMT) from VLDLR knockout (KO) mice into wild-type (WT) recipient mice attenuates insulin resistance in diet-induced obesity (DIO), simultaneously with reduced adipose tissue inflammation. Altogether, our data suggest that upregulated macrophage VLDLR could provoke insulin resistance by enhancing pro-inflammatory signaling pathways, accompanied with altered lipid profiles under lipid-rich conditions in obesity.

## Results

**VLDLR is highly expressed in ATMs from obese adipose tissue.** VLDLR is abundantly expressed in adipose tissue[28, 29]. However, it is largely unknown whether VLDLR expression might be altered in obese adipose tissue. To address this, VLDLR expression was examined in EATs from lean and obese mice. Compared to lean EATs, the level of VLDLR mRNA was elevated in obese EATs (Fig. 1a, b). mRNA levels of other LDLR family members, including LDLR and apolipoprotein receptor (ApoER) 2, were not significantly changed in EATs of obese mice (Fig. 1a, b). As positive controls, the mRNA levels of TNFα and Acrp30 were examined. In accordance with a previous report[36], the level of VLDLR mRNA in human adipose tissue showed a positive correlation with BMI (Supplementary Fig. 1). To further characterize the expression patterns of adipose tissue VLDLR, EATs were fractionated into adipocytes and SVCs. Unlike in adipocytes, the level of VLDLR mRNA was elevated in SVCs from HFD-induced obese mice as compared to those from control animals (Fig. 1c). To verify high expression of VLDLR in SVCs of DIO, SVCs were further separated into F4/80 and CD11b double-positive ATMs by using fluorescence-activated cells sorting. Upon HFD, the level of VLDLR mRNA was clearly raised in F4/80+ and CD11b+ ATMs, while those of LDLR and ApoER 2 were not altered (Fig. 1d). Moreover, elevated VLDLR protein was detected in CD11b+ ATMs from obese adipose tissues (Fig. 1e). In DIO, the levels of VLDLR mRNA were also elevated in peritoneal and liver macrophages (Supplementary Fig. 2a, b). On the other hand, the level of VLDLR mRNA in liver macrophages was quite low (Supplementary Fig. 2b). Together, these results indicated that VLDLR is highly expressed in obese adipose tissue, particularly in ATMs.

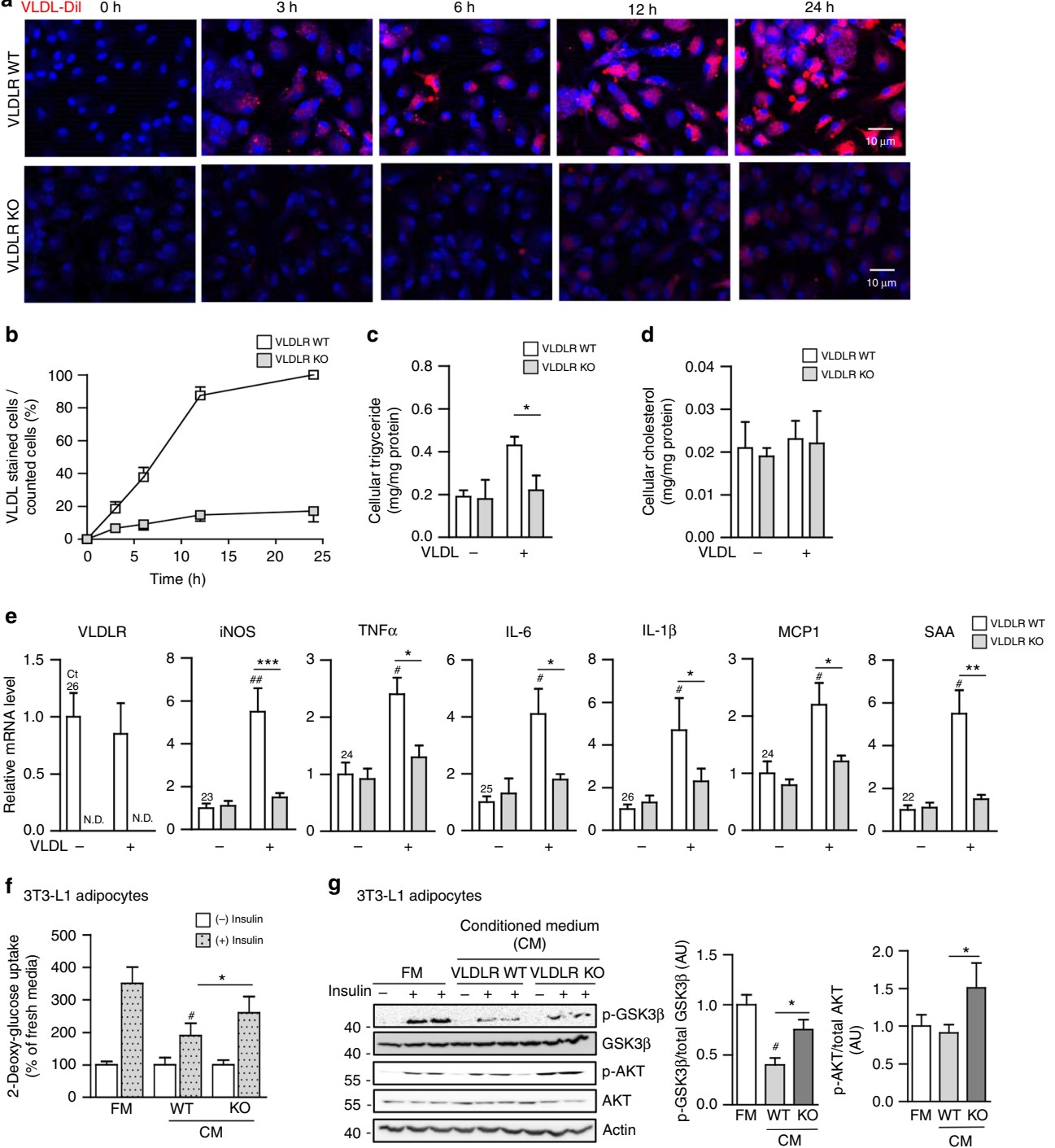

**Fig. 3** Macrophage VLDLR deficiency reduces intracellular triglyceride accumulation and pro-inflammatory gene expression upon VLDL. **a** Peritoneal macrophages from WT or VLDLR KO mice were treated with VLDL-Dil (10 μg/ml) for the indicated periods. VLDL-Dil (red) was detected by fluorescence microscopy. **b** Quantification of accumulated VLDL-Dil fluorescence in WT and VLDLR KO macrophages during different incubation periods. **c**, **d** Intracellular triglyceride (**c**) and cholesterol (**d**) in WT or VLDLR KO macrophages with (+) or without (−) human VLDL (30 μg/ml) challenges. **e** Relative mRNA levels of VLDLR and pro-inflammatory genes in WT or VLDLR KO macrophages with (+) or without (−) human VLDL (30 μg/ml) challenge. Each mRNA level was normalized to cyclophilin mRNA. **c**–**e** Data represent mean ± SD. #$P < 0.05$, ##$P < 0.01 ±$ VLDL; *$P < 0.05$, **$P < 0.01$, ***$P < 0.001$, Student's $t$-test. **f**, **g** Insulin-dependent glucose uptake ability (**f**) and insulin signaling cascades (**g**) were examined in 3T3-L1 adipocytes after conditioned media treatment from WT or VLDLR KO macrophages in the presence of human VLDL (30 μg/ml). FM and CM stand for fresh media and conditioned media, respectively. **f**, **g** Data represent mean ± SD. #$P < 0.05$ vs. FM, *$P < 0.05$, Student's $t$-test

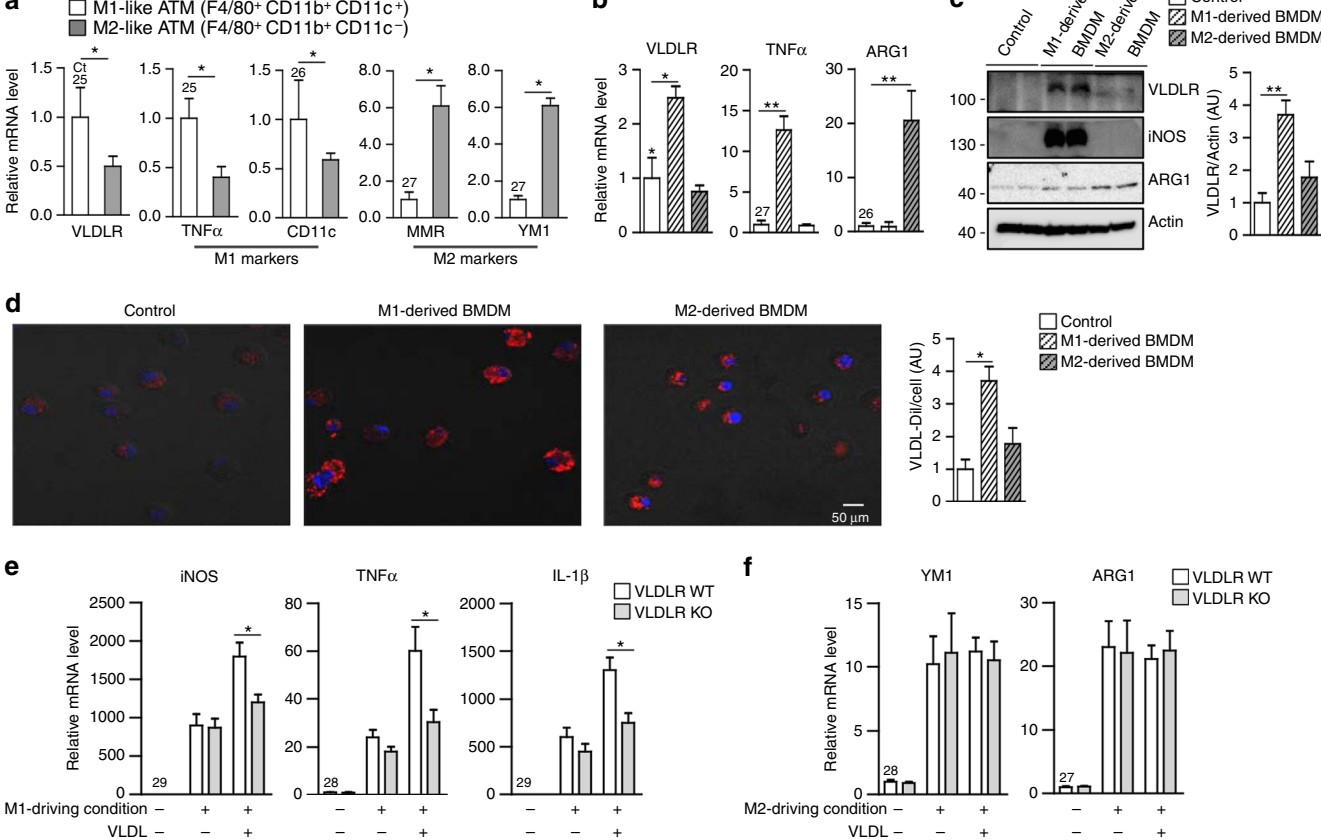

**Fig. 4** VLDLR accelerates M1-like macrophage polarization with VLDL. **a** Relative mRNA level of VLDLR in sorted M1-like macrophages (F4/80[+], CD11b[+], CD11c[+]) and M2-like macrophages (F4/80[+], CD11b[+], CD11c[−]) from EATs ($n = 9$). Relative levels of VLDLR mRNA (**b**) and protein (**c**) in BMDMs cultured under M1-driving (LPS and IFN-γ) or M2-driving (IL-4) condition for 24 h. **d** Intracellular accumulation of VLDL-Dil (red) in BMDMs after M1-driving or M2-driving condition. **e**, **f** Relative mRNA levels of pro-inflammatory (**e**) or anti-inflammatory (**f**) genes in M1-derived or M2-derived BMDMs from WT and VLDLR KO mice with or without human VLDL (30 μg/ml). Each mRNA level was normalized to cyclophilin mRNA. Data represent mean ± SD. *$P <$ 0.05, **$P <$ 0.01, Student's $t$-test

**VLDLR-overexpressed macrophages potentiate pro-inflammatory responses in the presence of VLDL.** To investigate whether macrophage VLDLR might contribute to storage of intracellular lipid metabolites, VLDLR was overexpressed in peritoneal macrophages (Fig. 2a, b). As shown in Fig. 2c, d, the level of intracellular triglycerides was increased in VLDLR-overexpressing macrophages in the presence of VLDL, while that of cholesterol was not altered. Given the high expression of VLDLR in obese ATMs, we next tested whether VLDLR in ATMs might be involved in pro-inflammatory responses. In VLDLR-overexpressing macrophages, the presence of VLDL stimulated the expression of pro-inflammatory marker genes, including iNOS, TNFα, monocyte chemoattractant protein (MCP)-1, serum amyloid A (SAA), IL-1β, and interferon (IFN)γ (Fig. 2e). These results suggested that elevation of macrophage VLDLR expression would stimulate pro-inflammatory responses in the presence of VLDL, simultaneously with intracellular triglycerides accumulation.

**Macrophage VLDLR deficiency attenuates VLDL-induced pro-inflammatory responses.** As VLDLR-overexpressing macrophages had elevated intracellular triglycerides in the presence of VLDL (Fig. 2), we investigated whether macrophages could uptake VLDL via VLDLR. To address this, peritoneal macrophages isolated from WT or VLDLR KO mice were challenged with fluorescence-conjugated VLDL (VLDL-DiI). As

illustrated in Fig. 3a, b, VLDLR KO macrophages hardly took up VLDL compared to WT macrophages. While WT macrophages accumulated intracellular triglycerides with VLDL in a time-dependent manner, VLDLR KO macrophages marginally increased intracellular triglycerides (Fig. 3c). Intracellular cholesterol did not differ between WT and VLDLR KO macrophages with or without VLDL (Fig. 3d). It has been reported that VLDL would be uptaken by receptor-mediated endocytosis or lipoprotein lipase (LPL)-dependent lipolysis[28, 29]. To test whether LPL might be involved in VLDL uptaking, we investigated LPL expression and its enzymatic activity in WT and VLDLR KO macrophages. As shown in Supplementary Fig. 3, the levels of LPL mRNA and its enzymatic activity in macrophages were not associated with VLDLR expression. Moreover, suppression of LPL via siRNA did not significantly alter cellular triglycerides and cholesterol contents in the absence or presence of VLDL. To validate the potential roles of macrophage VLDLR, peritoneal macrophages from WT or VLDLR KO mice were treated with or without VLDL and subjected to gene expression profiling. In the presence of VLDL, VLDLR KO macrophages did not have augmented expression of various pro-inflammatory marker genes, such as iNOS, TNFα, IL-6, IL-1β, MCP-1, and SAA, while WT macrophages did show stimulated expression of these pro-inflammatory genes (Fig. 3e). These results imply that macrophages would uptake VLDL via VLDLR and potentiate inflammatory responses, concomitantly with intracellular triglycerides accumulation. It has been well established that cytokines

produced from macrophages could impair insulin action in adipocytes[16, 17]. Thus, we speculated that deficiency of macrophage VLDLR might affect insulin-induced glucose uptake and

insulin signaling in adipocytes. To address this, conditioned media (CM) were collected from VLDL-treated peritoneal macrophages isolated from WT or VLDLR KO mice and were treated

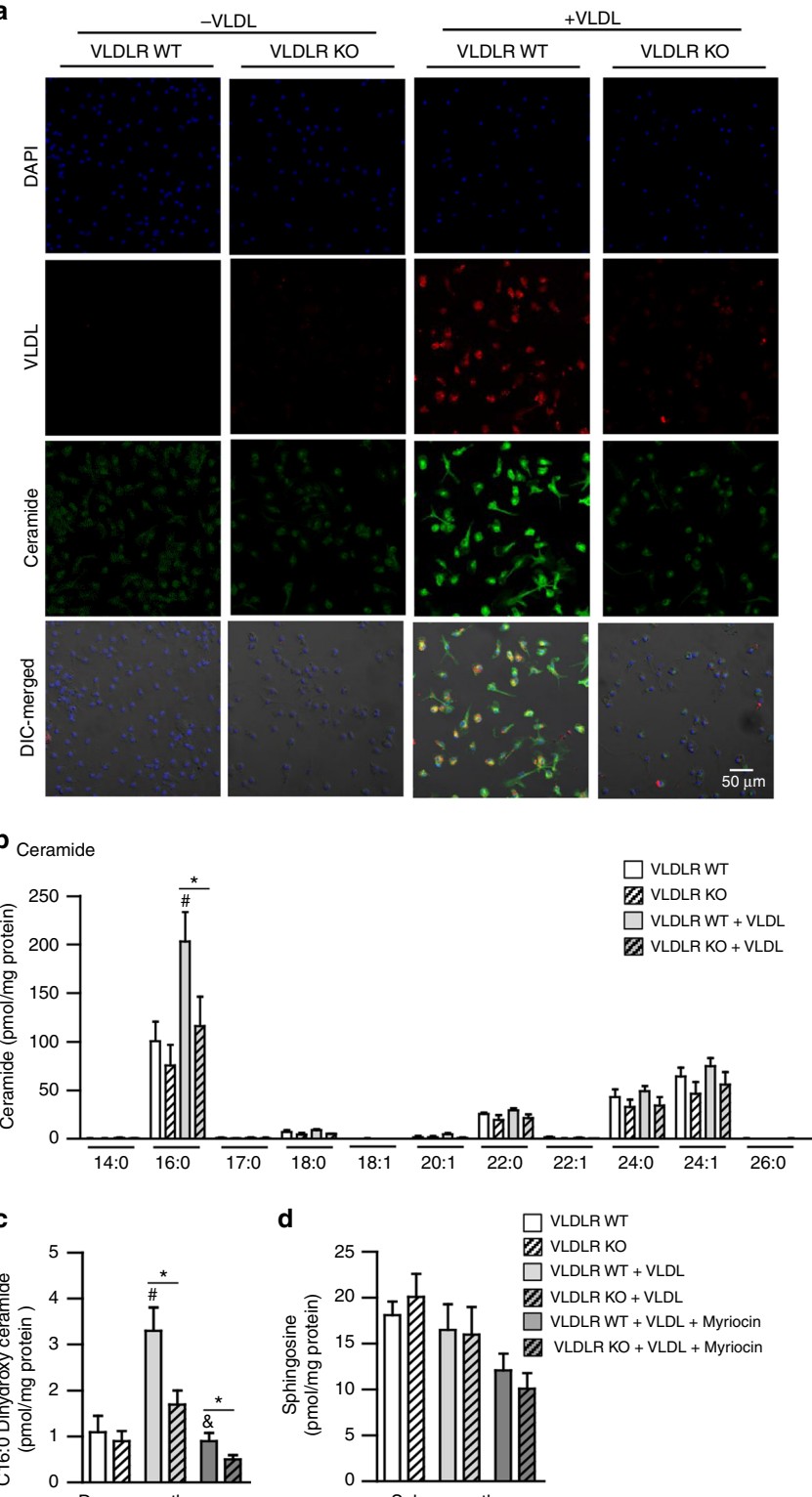

**Fig. 5** The level of C16:0 ceramides is elevated through VLDL–VLDLR axis in macrophages. **a** Microscope imaging analysis of the nucleus (blue), VLDL-Dil (red), ceramides (green), and DIC-merged of WT or VLDLR KO BMDMs with or without VLDL (30 μg/ml). **b** Lipid profile analysis was performed in BMDMs from WT or VLDLR KO mice with or without VLDL incubation. **c, d** The levels of C16:0 dihydroxy ceramides and sphingosines were measured in WT or VLDLR KO BMDMs with or without VLDL (30 μg/ml) or myriocin (10 μM). Data represent mean ± SD. $^{\#}P < 0.05$ vs. VLDL, $^{\&}P < 0.05$ vs. myriocin; $^{*}P < 0.05$, Student's *t*-test

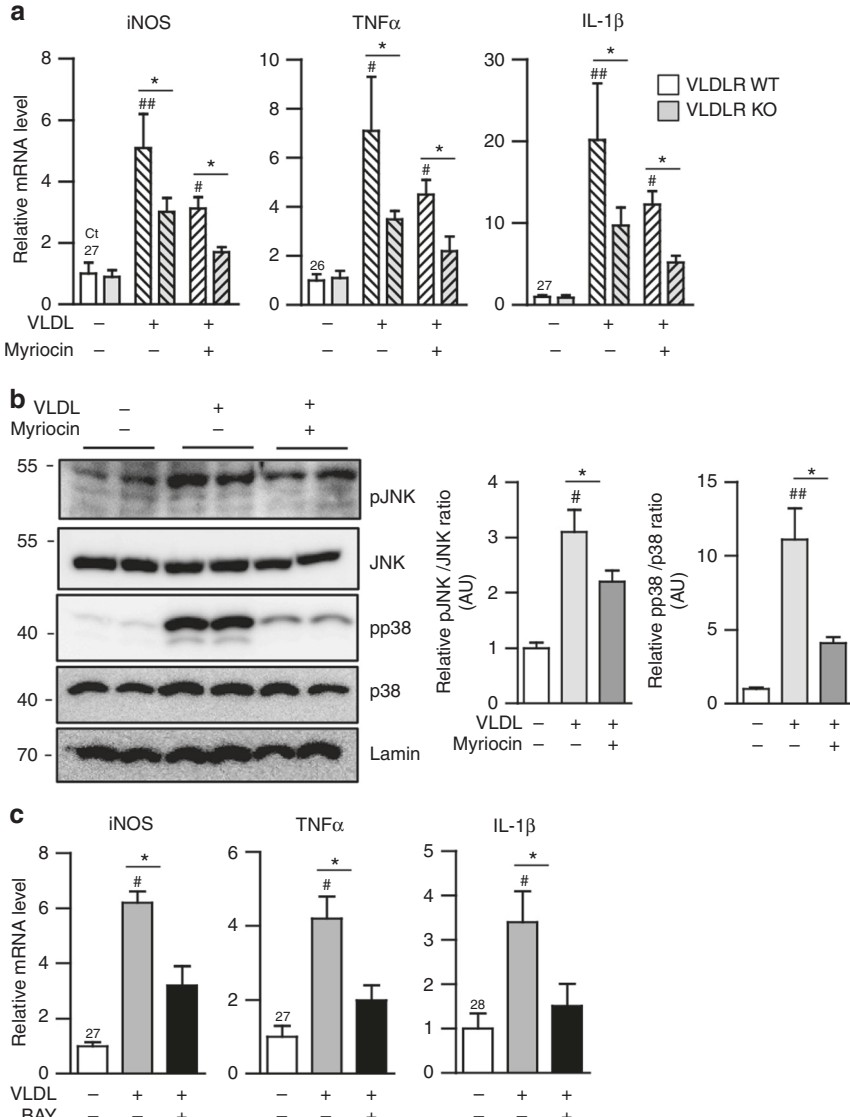

**Fig. 6** MAPK pathway is involved in VLDL-induced inflammatory responses through ceramides. **a** Relative mRNA levels of pro-inflammatory genes in BMDMs from WT and VLDLR KO mice with human VLDL (30 μg/ml) or myriocin (10 μM) treatment. **b** Western blots of MAPK (JNK and p38) phosphorylation in BMDMs treated with or without human VLDL (30 μg/ml) or myriocin (10 μM). **c** Relative mRNA level of pro-inflammatory genes in BMDMs cultured with or without human VLDL (30 μg/ml) or BAY 11-7082 (10 μM). Each mRNA level was normalized to cyclophilin mRNA. Data represent mean ± SD. #$P < 0.05$ ± VLDL; ##$P < 0.01$ ± VLDL; *$P < 0.05$, Student's $t$-test

to differentiated 3T3-L1 adipocytes. Compared with CM from VLDL-treated WT macrophages, CM from VLDL-treated VLDLR KO macrophages slightly but substantially enhanced insulin-stimulated glucose uptake ability and increased the level of glucose transporter 4 (GLUT4) mRNA (Fig. 3f and Supplementary Fig. 4). Furthermore, the phosphorylation levels of AKT and GSK3β were elevated in adipocytes treated with CM from VLDLR KO macrophages upon insulin (Fig. 3g). Together, these results indicated that macrophage VLDLR could mediate pro-inflammatory responses in the presence of VLDL, which would aggravate insulin action in adipocytes.

**VLDLR accelerates M1-like macrophage polarization upon VLDL.** The findings that macrophage VLDLR expression was upregulated in obese adipose tissue and stimulated pro-inflammatory gene expression upon VLDL promoted us to test whether VLDLR might be abundantly expressed in either

M1- or M2-like ATMs. ATMs were fractionated into M1-like ATMs (F4/80+, CD11b+, and CD11c+) and M2-like ATMs (F4/80+, CD11b+, and CD11c−). Compared with M2-like ATMs, M1-like ATMs more abundantly expressed VLDLR mRNA as well as TNFα and CD11c mRNAs (Fig. 4a). To gain further insights in the role of macrophage VLDLR, LPS, and IFNγ or IL-4, were added to cultured BMDMs to induce M1- or M2-like macrophage polarization, respectively. In BMDMs, VLDLR mRNA and protein were upregulated under M1-like macrophage-driving condition rather than under M2-like macrophage-driving condition (Fig. 4b, c). As positive controls, TNFα and arginase 1 (ARG1) expression were measured for M1-driving and M2-driving conditions, respectively (Fig. 4b, c). As VLDL was uptaken through macrophage VLDLR (Fig. 3), VLDL-DiI was incubated in BMDMs during M1- or M2-like macrophage polarization. As shown in Fig. 4d, M1-derived BMDMs accumulated more VLDL-DiI than did M2-derived BMDMs, implying that M1-like macrophages would uptake and store more

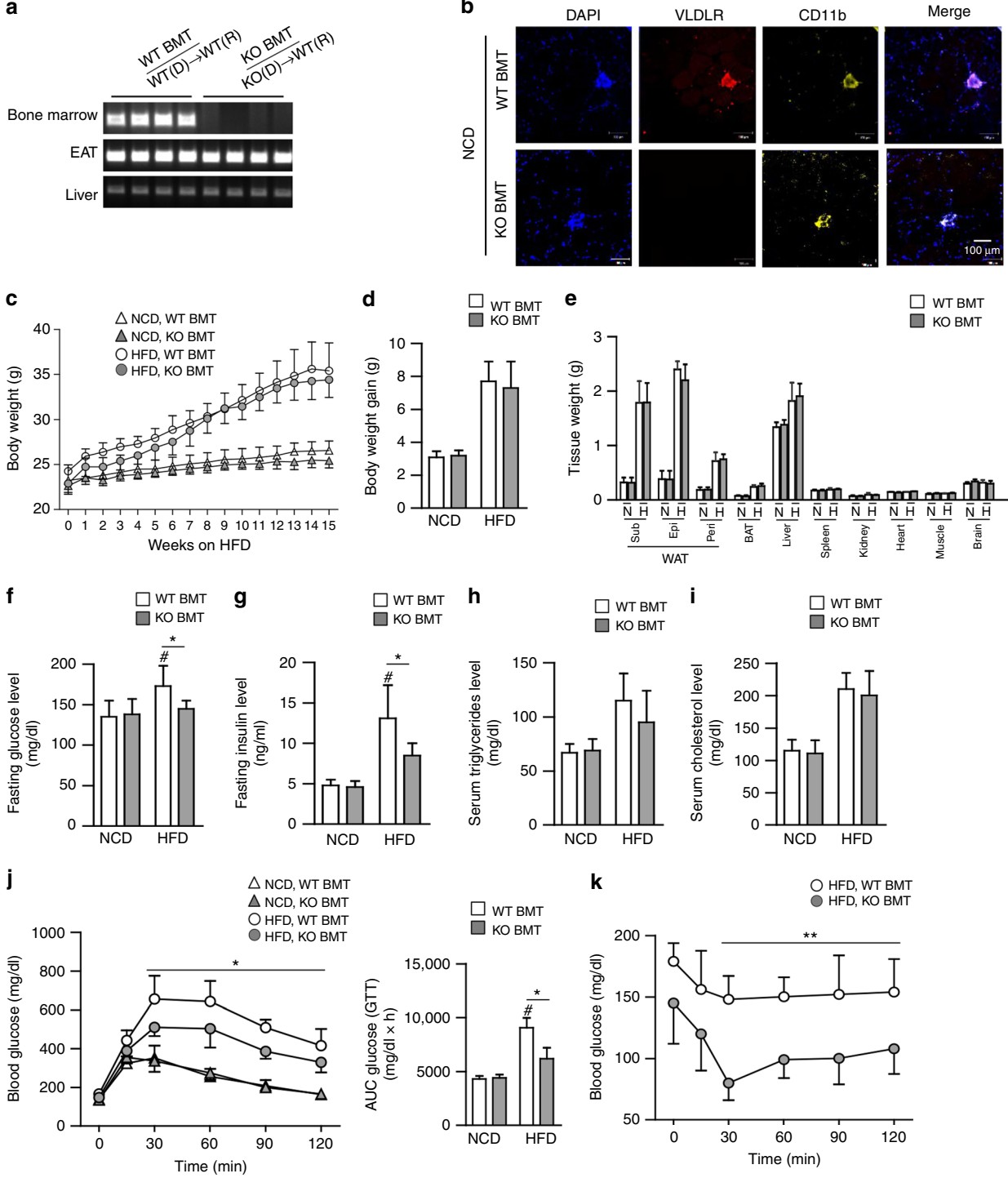

**Fig. 7** HFD-fed KO BMT mice ameliorate insulin resistance. **a–k** WT BMT [WT(D)→WT(R)] and KO BMT [KO(D)→WT(R)] mice ($n = 6$/group) were fed NCD or HFD for 15 weeks. **a** WT and KO BMT mice were genotyped. Genomic DNAs were isolated from bone marrow, EATs, and livers from WT and KO BMT mice. **b** Expression patterns of VLDLR protein in ATMs of WT and KO BMT mice. Whole-mount immunohistochemistry of the nucleus (blue), VLDLR (red), and CD11b (yellow) in NCD-fed EAT. **c** Body weights throughout the experimental period. **d** Total body weight gain and **e** various tissue weights in WT and KO BMT mice after NCD (N)-fed or HFD (H)-fed 15 weeks. **f–i** Fasting serum glucose (**f**), insulin (**g**), triglyceride (**h**), and cholesterol (**i**) in WT and KO BMT. **j** Glucose tolerance test (GTT), area under the curve (AUC), and **k** insulin tolerance test (ITT) analysis of WT and KO BMT mice. Data represent mean ± SD. #$P < 0.05$ vs. HFD, *$P < 0.05$; **$P < 0.01$, Student's $t$-test

VLDL due to elevated VLDLR expression. Next, to validate whether VLDLR might contribute to promote M1-like macrophage polarization in the presence of VLDL, BMDMs from WT or VLDLR KO mice were induced to M1- or M2-like macrophage phenotype with or without VLDL incubation. In the presence of VLDL, M1-derived BMDMs from WT mice further increased the expression of M1 marker genes, such as iNOS, TNFα, and IL-1β (Fig. 4e). On the contrary, in M2-derived BMDMs from either WT or VLDLR KO mice, the mRNA levels of M1 and M2 marker genes were not altered with or without VLDL (Fig. 4f and

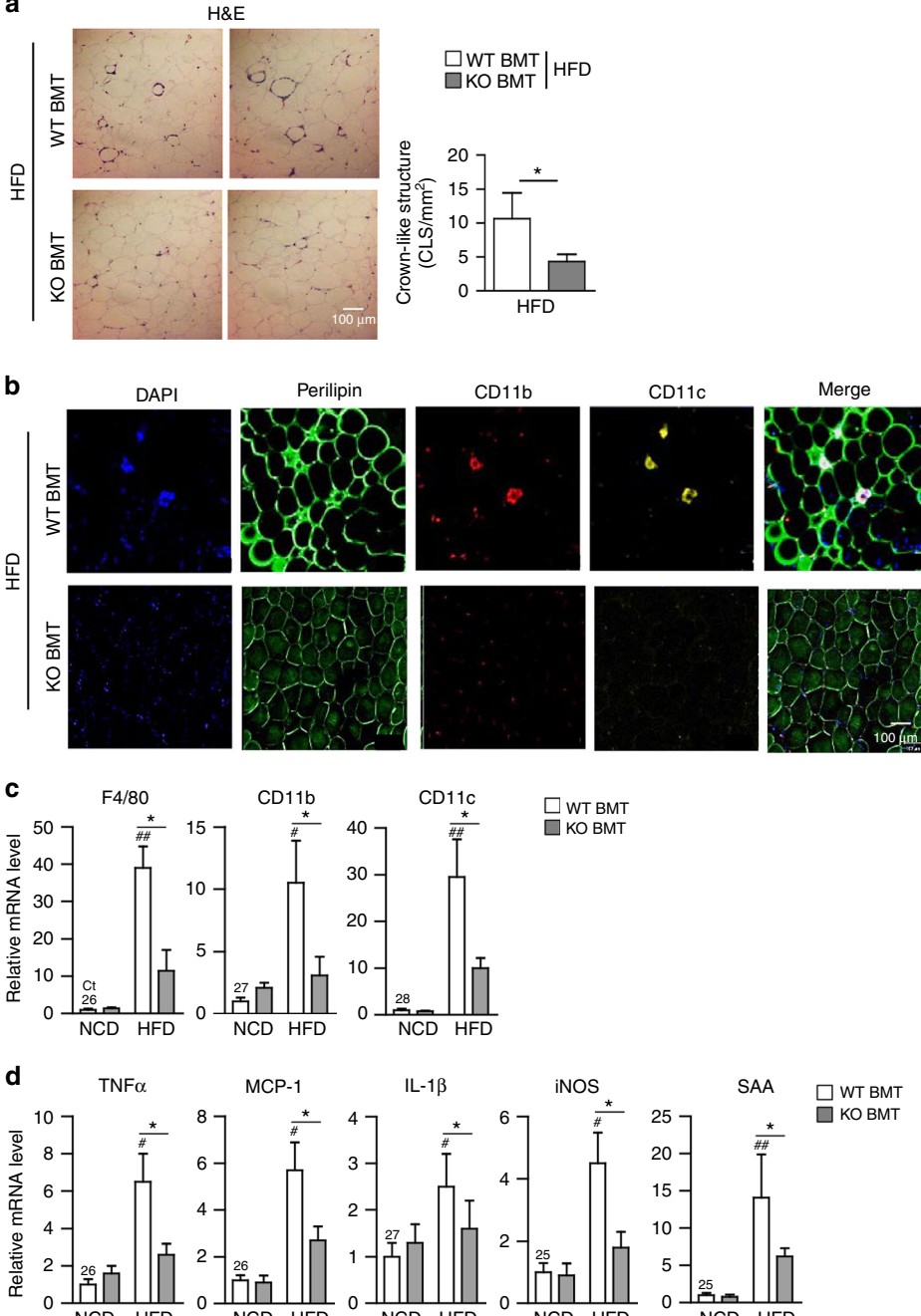

**Fig. 8** Adipose tissue inflammation is reduced in HFD-fed KO BMT mice. **a** Histological analysis of EAT from WT and KO BMT mice using H&E staining. **b** Whole-mount immunohistochemistry of the nucleus (blue), CD11b (red), and CD11c (yellow). Immune cell markers in EATs from WT and KO BMT mice. **c**, **d** Relative mRNA levels of macrophage (**c**) and pro-inflammatory (**d**) markers in EATs from WT and KO BMT mice. Each mRNA level was normalized to cyclophilin mRNA. Data represent mean ± SD. #$P < 0.05$ vs. HFD; ##$P < 0.01$ vs. HFD; *$P < 0.05$, Student's $t$-test

Supplementary Fig. 5a). These data suggested that macrophage VLDLR could potentiate M1-like macrophage polarization by uptaking VLDL.

**The level of C16:0 ceramides is elevated in VLDL-treated macrophages through VLDLR.** It has been reported that lipid metabolites of VLDL could be converted into various ceramides, and increased ceramides promote pro-inflammatory responses in macrophages[37–39]. To investigate whether the VLDL-VLDLR axis might affect ceramide metabolism in macrophages, the levels of intracellular ceramides were determined in WT or VLDLR KO

macrophages with or without VLDL challenges. As shown in Fig. 5a, VLDLR KO macrophages accumulated less ceramides than did WT macrophages with VLDL incubation, which was positively correlated with the level of intracellular VLDL. Although it has been shown that elevated circulating ceramides would confer systemic insulin resistance[40, 41], the levels of secreted ceramides were not different in CM from WT and VLDLR KO BMDMs (Supplementary Fig. 6). Next, to identify specific ceramide species in VLDL-incubated macrophages, lipidomic profiling was conducted using WT or VLDLR KO macrophages. In WT macrophages, the level of C16:0 ceramides was elevated by VLDL, whereas VLDLR KO macrophages did not

increase C16:0 ceramides in the presence of VLDL (Fig. 5b). To further elucidate how VLDL-dependent C16:0 ceramides might be accumulated in WT macrophages, we measured the levels of C16:0 dihydroxy ceramides for the de novo pathway and sphingosine for the salvage pathway. In VLDL-incubated WT macrophages, elevated level of C16:0 dihydroxy ceramides was decreased by myriocin, which is a potent inhibitor of de novo synthesis of ceramides (Fig. 5c). However, the level of sphingosine was not altered by myriocin in WT or VLDLR KO macrophages (Fig. 5d). These data implied that macrophage VLDLR would increase intracellular C16:0 ceramide species in the presence of VLDL, at least probably, via de novo synthesis.

**Increased ceramides promote pro-inflammatory responses via VLDL-VLDLR axis in macrophages.** It has been demonstrated that accumulated VLDL stimulates pro-inflammatory gene expression in macrophages[34, 35]. However, underlying mechanisms by which VLDL could induce pro-inflammatory responses are not fully understood in macrophages. To test whether VLDL-VLDLR axis-dependent ceramide accumulation might mediate pro-inflammatory responses, pro-inflammatory gene expression was examined in WT or VLDLR KO macrophages under various conditions. In the presence of VLDL, the levels of iNOS, TNFα, and IL-1β mRNA were less increased in VLDLR KO macrophages than WT macrophages (Fig. 6a). Moreover, myriocin downregulated the expression of pro-inflammatory genes in either WT or VLDLR KO macrophages treated with VLDL (Fig. 6a). Given that VLDLR would be involved in the clearance of triglycerides derived from VLDL and chylomicrons[42], we investigated whether chylomicron might affect inflammatory responses and cellular ceramides in macrophages. Unlike VLDL, the levels of pro-inflammatory cytokine gene expression and intracellular ceramides in macrophages were not altered by chylomicron (Supplementary Fig. 7). On the other hand, it has been well established that mitogen-activated protein kinase (MAPK) signaling cascades play a crucial role in the regulation of pro-inflammatory gene expression upon various cellular stresses[14, 15]. Thus, to understand the molecular mechanisms by which macrophage VLDLR could upregulate pro-inflammatory gene expression with ceramides, several MAPK signaling pathways were tested. As shown in Fig. 6b, the phosphorylation levels of c⁻ Jun N-terminal kinase (JNK) and p38 MAPK were potentiated in VLDL-treated macrophages. However, myriocin attenuated their phosphorylation, even in the presence of VLDL (Fig. 6b). Given that nuclear factor (NF)-κB is one of the key transcription factors governing the expression of most pro-inflammatory genes, we tested its involvement in the regulation of pro-inflammatory genes in VLDL-treated macrophages. As indicated in Fig. 6c, BAY 11-7082, an inhibitor of IKKα, decreased the levels of VLDL-induced iNOS, TNFα, and IL-1β mRNA in VLDL-treated macrophages. These data suggested that the VLDL-VLDLR axis in macrophages would stimulate, at least partly, the MAPK and NF-κB pathways, leading to the promotion of pro-inflammatory responses upon increased cellular ceramides.

**Deficiency of hematopoietic VLDLR improves glucose intolerance and insulin intolerance in DIO.** Given that macrophages from VLDLR KO mice attenuated pro-inflammatory responses, even in the presence of VLDL, we asked whether deficiency of macrophage VLDLR might relieve adipose tissue inflammation in DIO. To address this, BMT experiments were performed via adoptive transfer of bone marrow cells from VLDLR KO mice or their WT littermates into lethally irradiated C57BL/6 WT mice.

Then, both groups were fed a HFD, and we compared the WT mice that had received bone marrow cells from VLDLR KO mice (KO BMT) with those that had received WT bone marrow (WT BMT). Genotyping analysis confirmed that bone marrow cells in KO BMT mice were successfully replaced with VLDLR KO bone marrow cells (Fig. 7a). Immunohistological analysis revealed that VLDLR was highly expressed in CD11b⁺ ATMs of WT BMT mice, whereas it was hardly detected in KO BMT mice (Fig. 7b). Upon HFD, there were no significant differences in body weights and body weight gains between WT BMT and KO BMT mice (Fig. 7c, d). In addition, there were no changes in organ weight in brown, epididymal, subcutaneous, and perirenal adipose tissues, and other metabolic organs (Fig. 7e). Nonetheless, interestingly, fasting glucose and insulin were decreased in KO BMT mice as compared to WT BMT mice after HFD (Fig. 7f, g). Moreover, HFD-fed KO BMT mice were more glucose-tolerant and insulin-sensitive than HFD-fed WT BMT mice (Fig. 7j, k). However, the serum levels of triglycerides and cholesterols were not different between both groups (Fig. 7h, i). These data implied that the reconstitution of hematopoietic cells in WT mice with VLDLR KO bone marrow cells would ameliorate insulin resistance in DIO.

**Deficiency of macrophage VLDLR alleviates adipose tissue inflammation in DIO.** To investigate whether improved glucose intolerance and insulin resistance in HFD-fed KO BMT mice might be associated with decreased adipose tissue inflammation, macrophage infiltration in adipose tissues of HFD-fed WT BMT and KO BMT mice was determined. Compared to HFD-fed WT BMT mice, the number of crown-like structures (CLSs) was decreased in HFD-fed KO BMT mice (Fig. 8a). Consistent with the decreased number of CLSs, HFD-fed KO BMT showed less accumulation of CD11c⁺ pro-inflammatory macrophages than HFD-fed WT BMT mice (Fig. 8b). Furthermore, mRNA levels of macrophage marker genes, such as F4/80, CD11b, and CD11c, were lower in EATs of HFD-fed KO BMT mice as compared to those of HFD-fed WT BMT mice (Fig. 8c). Given that increased accumulation of ATMs could be attributable to several cellular events, including infiltration, retention, and proliferation in DIO[43–45], we have tested these possibilities by injecting GFP-positive monocytes into clodronate-induced phagocytic cell-deficient WT recipient mice that were adoptively transferred with WT or VLDLR KO bone marrow cells (Supplementary Fig. 8a). As shown in Supplementary Fig. 8b–e, it is likely that hematopoietic VLDLR deficiency would primarily affect macrophage infiltration into obese adipose tissue. Additionally, mRNA levels of pro-inflammatory genes, such as TNFα, MCP-1, IL-1β, iNOS, and SAA, were lower in EATs and subcutaneous adipose tissue of HFD-fed KO BMT mice (Fig. 8d and Supplementary Fig. 9a). On the contrary, there was no significant difference in expression level of macrophage marker and inflammatory genes in liver (Supplementary Fig. 9b). Together, these data indicated that macrophage VLDLR could potentiate adipose tissue inflammation in DIO.

## Discussion

Elevated plasma triglyceride-rich lipoprotein, such as VLDL, is considered a risk factor for prevalence of obesity, type 2 diabetes mellitus, and atherosclerosis[46–48]. As peripheral VLDLR is involved in the clearance of circulating VLDL and/or chylomicron[28, 29, 42], dysregulation of VLDLR has been implicated in metabolic complications[30–35]. In accordance, VLDLR-deficient mice are protective from DIO and improve systemic insulin resistance[32, 33]. Recently, Nguyen et al. reported that VLDLR KO mice attenuated adipocyte hypertrophy and that ER

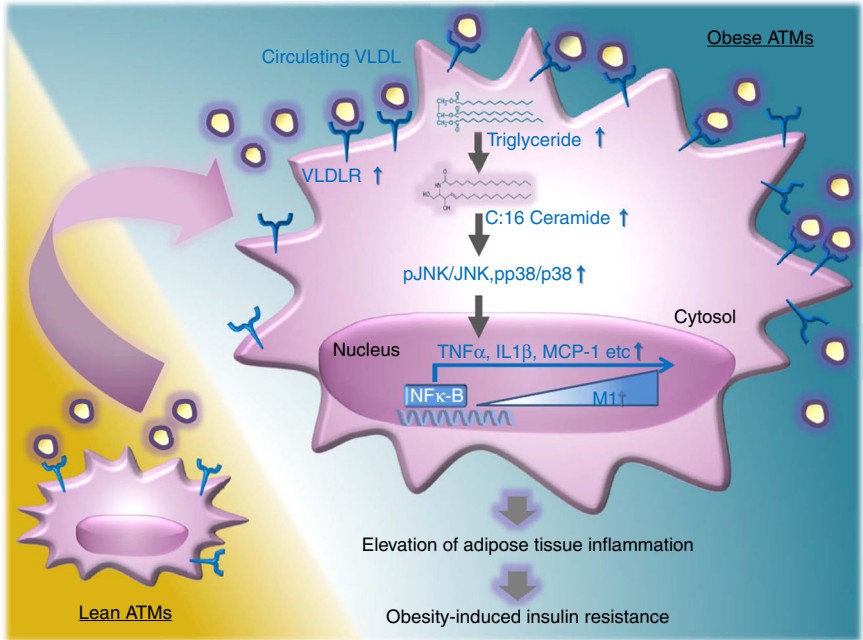

**Fig. 9** Working model. In obese ATMs, VLDL–VLDLR axis stimulates adipose tissue inflammation, accompanied with obesity-induced insulin resistance

stress and pro-inflammatory responses were downregulated in adipose tissue upon HFD[33]. However, it is largely unknown whether macrophage VLDLR might be crucial for obesity-induced insulin resistance through adipose tissue inflammation. Here we have shown that elevated VLDLR in ATMs would be a key factor to provoking adipose tissue inflammation and insulin resistance in obesity (Fig. 9).

Emerging evidence has indicated that macrophage polarization is important in the progress of metabolic disorders[13–18, 49]. For example, M1-like macrophage polarization contributes to metabolic impairment that is closely associated with obesity and diabetes[12–15]. Moreover, deletion of M1-like macrophages ameliorates insulin resistance in obese mice, whereas the reduction of M2-like macrophages predisposes lean mice to the development of insulin resistance[50, 51]. Thus, it is important to elucidate the molecular mechanisms by which macrophages could assume either M1- or M2-like character, which affects metabolic alterations in obesity. In this study, several lines of evidence supported the idea that macrophage VLDLR would exert regulatory roles in M1-like macrophages in obese adipose tissue. First, M1-like macrophages highly expressed VLDLR as compared to M2-like macrophages in adipose tissue. Second, M1-derived BMDMs contained more ceramides than M2-derived BMDMs in the presence of VLDL (Supplementary Fig. 5b). Third, VLDL stimulated pro-inflammatory gene expression in VLDLR-overexpressing macrophages. Fourth, VLDLR-deficient macrophages showed lower pro-inflammatory responses in the presence of VLDL, eventually leading to augmented insulin action in adipocytes. Lastly, VLDLR expression was elevated during M1-derived macrophage differentiation. In addition, macrophages from VLDLR KO mice had lower expression of pro-inflammatory genes during M1-derived macrophage differentiation in the presence of VLDL. These data imply that macrophage VLDLR would be important for potentiating M1-like macrophage polarization in the presence of VLDL. Given that the balance shifting between M1- and M2-like macrophages is critical for the progression of inflammatory responses in obese adipose tissue, it is plausible that macrophage VLDLR elevation in obesity confers systemic insulin resistance through adipose tissue inflammation.

Ceramides are a member of the sphingolipid family and are essential components of the cell membrane[52–54]. Recent studies have shown that increased ceramide level in obese adipose tissue coincides with development of insulin resistance[53–56]. In addition, pharmacological inhibition or genetic ablation of ceramide synthesis protects from an increase in adiposity and improves systemic insulin sensitivity in obesity[57–60]. Thus, the regulation of ceramide production has been proposed as a potential therapeutic target against obesity-mediated metabolic diseases[53–60]. Since macrophage VLDLR plays an important role in the uptake of lipid metabolites in VLDL[37, 38], we assessed whether macrophage VLDLR deficiency might change certain lipid metabolites in the presence of VLDL. In macrophages, the level of ceramides was increased, probably, via VLDL-VLDLR axis. Furthermore, analysis of the sphingolipid composition revealed that the levels of C16:0 ceramides and their precursors (C16:0 dihydroxy ceramides) were selectively reduced in VLDLR KO macrophages, whereas ceramide species with other chain lengths were not altered. De novo synthesis of C16:0 ceramides contributes to inflammasome activation in macrophages and is a key component that connects lipid oversupply and inflammatory pathways as well as insulin resistance in obesity[58–60]. In accordance with these findings, we observed that inhibition of ceramide synthesis could suppress inflammatory signaling pathways, which consequently downregulated the MAPK pathways and NF-κB signaling in VLDL-treated macrophages. These data indicate that C16:0 ceramide synthesis via VLDL-VLDLR axis in macrophages would stimulate pro-inflammatory responses, at least partly, via MAPK pathways. As it has been reported that MAPK activation in macrophages would promote intracellular ceramide contents[61], we tested whether MAPK pathways might affect the level of intracellular ceramides upon VLDL. As shown in Supplementary Fig. 10, the levels of cellular ceramides in VLDL-treated macrophages were not altered in the absence or presence of MAPK inhibitors. Although it needs to be elucidated how cellular VLDL could be converted into C16:0 ceramides in macrophages, our data suggest that macrophage VLDLR could participate in alteration of certain sphingolipids via VLDL uptaking, which eventually leads to insulin resistance in obesity.

It has been reported that VLDLR is involved in inflammatory responses in immune cells[32, 33, 62, 63]. For instance, monocyte-derived macrophages from VLDLR-deficient mice exhibit reduced secretion of inflammatory cytokines[33–35]. Furthermore, VLDLR deficiency impairs IL-1β induction upon activation of transcription factor activator protein (AP)-1, which is known to be regulated by MAPK p38 in macrophages[34, 35]. In this study, we found that VLDLR-deficient macrophages showed down-regulation of pro-inflammatory responses and M1-like polarization; this effect would eventually restore insulin signaling in adipocytes. Notably, we discovered that deletion of VLDLR in hematopoietic cells would be sufficient to alleviate systemic insulin resistance in DIO. Furthermore, hematopoietic VLDLR deficiency alleviated pro-inflammatory cytokine gene expression and decreased macrophage accumulation in adipose tissue upon HFD. Also, we observed that ceramide contents in ATMs of HFD-fed KO BMT mice were decreased (Supplementary Fig. 11). Together, these findings suggest that VLDLR in hematopoietic cells would play a major role in mediating inflammatory responses under nutrient-rich stress, such as HFD. In future, it remains to be investigated whether the reduced adipose tissue inflammation in HFD-fed KO BMT mice might be also attributable to other hematopoietic immune cells such as lymphocytes, neutrophils, or eosinophils, as well as macrophages with VLDLR deficiency.

In conclusion, we have demonstrated that macrophage VLDLR plays important roles in mediating chronic inflammation and insulin resistance in DIO. In macrophages, ablation of VLDLR reduced cellular ceramides and relieved inflammatory responses in the presence of VLDL. Moreover, decreased pro-inflammatory signaling in VLDLR-ablated macrophages could prevent systemic insulin resistance in obesity (Fig. 9). Collectively, our data suggest that regulation of macrophage VLDLR would be one of potential therapeutic targets in obesity-induced metabolic disorders.

## Methods

**Animals and treatments**. All animal experiments were approved by the Seoul National University Animal Experiment Ethics Committee (SNU-130611-2). C57BL6/J and VLDLR-deficient mice were obtained from The Jackson Laboratory (Bar Harbor, ME; strain 002529 B6; 127S7-Vldlr<tm1Her>J) and housed in colony cages under 12-h light/12-h dark cycles. All mice were genotyped by PCR[64] and VLDLR-heterozygous mice were bred to generate WT and VLDLR-deficient littermates. Eight-week-old male mice were maintained on a normal chow diet (Zeigler, Gardners, PA) or 60% high-fat diet (Research Diet, New Brunswick, NJ). For intraperitoneal glucose tolerance test, NCD- or HFD-fed mice were fasted for 6 h, blood samples were collected, glucose was administered (2 g/kg body weight), and blood samples were drawn at 15, 30, 60, 90, and 120 min to measure plasma glucose level. For insulin tolerance test, HFD-fed mice were fasted for 3 h and then administered insulin (0.75 unit/kg body weight; Lilly, Indianapolis, IN). For BMT experiments, 8-week-old recipient male mice were lethally irradiated two times (total 5 Gy) by means of a[137]Cs source at an interval of 4 h. After irradiation of the bone marrow cells of the recipient mice, these mice received a transplant of $5 \times 10^6$ bone marrow cells from 8-week-old VLDLR KO male mice or their WT littermates via tail-vein injection[65].

**Human samples**. Human study was conducted according to the Declaration of Helsinki and was approved by ethics committees of SNUBH (SNUBH IRB#B-1203/147-006, #A111218-CP02), and all subjects provided their written informed consent. Visceral fat tissues were obtained from patients with normal metabolic profiles in urological surgery and were irrigated with saline. Total RNA was isolated from human adipose tissues using the TRIzol reagent (Invitrogen, Waltham, MA) according to the manufacturer's protocol. cDNA was synthesized using an iScript cDNA Synthesis kit (Bio-Rad, Hercules, CA).

**Cell culture**. Mice were intraperitoneally injected with sterile thioglycolate solution (3 ml/mouse) to harvest peritoneal macrophages[66]. Harvested peritoneal macrophages were cultured in RPMI 1640 medium (HyClone, Logan, UT) with 10% fetal bovine serum (HyClone, Logan, UT), 100 U/ml penicillin, and 100 mg/ml streptomycin. Bone marrow was flushed from the femur and tibia, dispersed, and cultured in DMEM medium containing 10% FBS and 20% L929 CM for 7 days. To promote M1- or M2-derived macrophages, bone marrow-derived macrophages

(BMDMs) were stimulated with LPS (10 ng/ml) and IFNγ (100 ng/ml) for M1-driving condition, and IL-4 (20 ng/ml) for M2-driving condition. To test effects of lipoproteins, human VLDL (Kalen Biomedical, #770100, Germantown, MD), and chylomicron (BioVision, #7285-1000, Milipas, CA) were purchased. According to the manufacturer's information, purchased VLDL contains 1.1 mg/ml protein, and chylomicron is composed of 98% lipids and 2% protein.

**Immunohistochemistry**. Whole-mounted EATs were incubated with primary antibodies against VLDLR (1:500 dilution; Abcam, #92943, Cambridge, MA), ceramide (1:500 dilution; Sigma-Aldrich, #C8104, St. Louis, MO), perilipin (1:1000; Fitzgerald, #20R-PP004, Stow, MA), CD11b (1:1000), and CD11c (1:1000; eBioscience, #E01079, #4290718, San Jose, CA). After incubation with fluorescently labeled secondary antibodies (Thermo Fisher Scientific, Waltham, MA) and 4′,6-diamidino-2-phenylindole (DAPI; Vector Laboratory, Burlingame, CA) staining, samples were examined under a Zeiss LSM 700-NLO confocal microscope.

**Lipid profiling**. Measurement of triglycerides was performed according to the manufacture's protocol (INFINITY™, Thermo Scientific, Waltham, MA). For lipid profiling of macrophages, lipid metabolites were separated using an Agilent 1260 liquid chromatography system and a specific column (Thermo Scientific Accucore HILIC 100 × 2.1 mm; particle size 2.6 μm). Mobile phase A consisted of acetonitrile/water (95:5) with 10 mM ammonium acetate, pH 8.0, and mobile phase B consisted of acetonitrile/water (50:50) with 10 mM ammonium acetate, pH 8.0. For separation of mobile phases, a column was equilibrated with 100% mobile phase A, ramped to 20% mobile phase B in 5 min, and held for 5 min. The column was finally equilibrated with 100% mobile phase A for 5 min. Mass spectrometry analysis was performed using an Agilent 6430 triple quadruple mass spectrometer.

**Flow cytometry analysis**. After incubation with blocking antibodies, SVCs were stained with monoclonal antibodies for 12h against CD11b, F4/80, and CD11c (1:1000 dilution; eBioscience, #E01079, #E08392-1636, #4290718, San Jose, CA). For sorting CD11c+ (M1-like macrophages) and CD11c− (M2-like macrophages), FACS Aria II (BD Bioscience, San Jose, CA) was used.

**Western blot analysis**. Macrophages, 3T3-L1 adipocytes, and EATs were lysed with radioimmunoprecipitation assay buffer. The proteins were separated by sodium dodecyl sulfate polyacrylamide gel electrophoresis and transferred to polyvinylidene fluoride membranes. The blots were probed with following primary antibodies for 12h incubation: anti-VLDLR, anti-lamin B (1:500 dilution; Abcam, #92943, #65986, Cambridge, MA), anti-β-actin (1:2000 dilution; Sigma-Aldrich, #A5441, St.Louis, MO), anti-p-GSK3β (Ser 9), anti-p-JNK (1:1000 dilution; Cell Signaling Technology, #D85E12, #85E11, Denver, MA), anti-GSK3β, anti-p-p38 (1:1000 dilution; Merck Millipore, #07-1413, #MABS64, Germany), anti-JNK, anti-p38 (1:1000 dilution; Santa Cruz Biotechnology, #sc-7345, #sc-81621, Dallas, TX).

**Statistical analysis**. The data are presented from multiple independent experiments, and represent the mean and standard deviation (SD). P values were calculated by Student's t-test; $P < 0.05$ was regarded significantly.

**Data availability**. The data that support the findings of this study are available from the corresponding author on reasonable request.

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

## Acknowledgements

We thank the members of the Laboratory of Adipocyte and Metabolism Research for helpful discussion. We also like to thank Kevin Timothy Fridianto for helping to perform the lipidomic profiling analysis. This work was supported by the National Creative Research Initiative Program of the National Research Foundation (NRF) funded by the Korean government (the Ministry of Science, ICT & Future Planning, 2011-0018312). K. C.S., I.H., J.P., Y.J., and J.I.K. were supported by the BK21 program.

## Author contributions

K.C.S., S.S.C., and J.B.K. designed the project. K.C.S. executed most experiments. I.H., J. P., and Y.J. performed BMT experiment. J.C. and J.P.K. examined lipidomic profiling analysis. G.Y.L. and S.H.C. provided human adipose tissue samples. K.C.S., I.H., S.S.C., J. I.K., and J.B.K. prepared the manuscript.

## Additional information

**Competing interests:** The authors declare no competing financial interests.

