## [Peer Review file · Nature Communications]

Reviewers' comments:

Reviewer #1 (Remarks to the Author):

In this paper, the Authors show that M1-like macrophages in obese adipose tissue express higher levels of VLDLR than M2-like macrophages. They then go on to use various *in vitro* and *in vivo* studies to show that the VLDLR is necessary for VLDL-mediated proinflammatory responses in macrophages. They also use chimeric mouse technology to demonstrate an effect of VLDL receptor macrophage deficiency on glucose tolerance and insulin sensitivity. As such, these studies build on previous papers showing that VLDLR KO protects mice from diet-induced obesity and that VLDL treatment promotes macrophage inflammation. The mechanistic aspects of this study relate to MAP kinase signaling and ceramide 16:0. The mechanistic studies are interesting but a bit thin.

Specific questions.

1. What is VLDLR expression in other macrophages from obese mice such as IP Macs and liver macrophages?
2. While of interest, the Authors do not show whether MAP kinase activation is a result of ceramide synthesis, or a cause. These are straightforward studies and should be performed.
3. With respect to insulin resistance, do the Authors think that the ceramide synthesized within macrophages upon VLDL treatment can be released and taken up into insulin target cells causing insulin resistance.
4. What is the effect of VLDL treatment on M2 macrophages with respect to inflammation and ceramide pathways compared to the M1 macrophages, since the M2 macrophages still express about 50% of the VLDLR.
5. How do the Authors explain the decrease in adipose tissue macrophage numbers in the VLDLR KO chimeric mice? Something must be affecting chemotaxis, retention, or proliferation of macrophages.
6. In Figure 3G, it would be preferable to measure AKT phosphorylation as an indicator of insulin action.
7. In Figure 3F, do the Authors have any ideas or data to show what factors are causing the changes in glucose transport. The introduction and discussion on this point are pretty generic and a bit out of date and do not take into account new data on macrophage-released factors. As the Authors undoubtedly know, there is a substantial literature on M2 macrophages and beigeing/BAT, possibly mediated through NE release. How does this play into the Authors studies?
8. There might be a mistake in Figure 6, since panels A, (iNOS and TNF α) look to be identical.
9. In many places the Authors state that IL1 β causes insulin resistance. What is the evidence for this, since much data is contradictory on this point?
10. The Authors should consider their results in light of the recent paper from the Glass laboratory on macrophage lipidomics (Oisi, et. al. Cell Metab. 25:42-427).
11. When the Authors measure macrophage TG content, does their method allow them to discriminate between fatty acids conjugated to glycerol vs. free intracellular fatty acids? Many of the effects the Authors describe could be due to increased intracellular SFA levels. This is an important distinction to try and make. It might be that the VLDLR is simply an added process to allow fatty acids to accumulate in macrophages.
12. Is there any data on VLDL receptor in human obese fat? either from the Authors or in the literature?
13. In Figure 2E, have the Authors measured other inflammatory factors besides the four listed.
14. In their summary Figure 9, MAP kinase is not listed. Where do the Authors think this fits into this pathway?

Reviewer #2 (Remarks to the Author):

Kyung Cheul Shin and colleagues investigated the role of the VLDL receptor (VLDLR) in lipid uptake, inflammation and M1/M2 polarisation of mouse macrophages *in vitro*, and determined the effect of hematopoietic VLDLR-deficiency on adiposity, glucose tolerance and insulin resistance in mice *in vivo*. They show that isolated macrophages take up VLDL-derived lipids which is coupled to ceramide production, expression of cytokines (e.g. iNOS, TNF α , and MCP1) and markers of M1 polarization. Most of these effects were abrogated or largely inhibited by VLDLR-deficiency. Hematopoietic deficiency for VLDLR did not affect high fat diet (HFD)-induced weight gain or fat mass, but improves fasting glucose and insulin and improved tolerance to glucose and insulin. The authors nicely demonstrate that part of the previously reported beneficial metabolic effects of VLDLR-deficiency (refs 27 and 28) can thus be conferred by hematopoietic cells, which may well be macrophages. The experiments seem to be adequately performed and in general the manuscript reads well. If strengthened, the data will be of interest to a wide public. I do have some reservations and comments as outlined below.

Major comments:

1. Novelty of the findings. Nguyen et al (ref 28) previously showed that VLDLR-deficient macrophages, upon incubation with VLDL, accumulate less lipids, have lower TNF α and IL-6 expression and produce less IL-6 and MCP-1 protein *in vitro*. Also, Goudriaan et al (ref 27) has demonstrated that whole-body VLDLR-deficiency markedly improves glucose tolerance in mice. The present data are thus not unexpected, which may compromise novelty to some extent.
2. Mode of action of VLDLR in VLDL uptake. The authors quite consistently mention that the VLDLR is involved in the 'uptake of VLDL'. This is only one of two general views. Another view is that the VLDLR functions as a docking protein, ensuring binding of triglyceride (TG)-rich lipoproteins (TRL) to the cell surface in vicinity of lipoprotein lipase (LPL; the expression pattern of both proteins is similar), allowing LPL to selectively delipidate TRL. The latter mechanism may indeed explain that TRL selectively donate TG-derived fatty acids to metabolically active tissues, while the uptake of TRL-derived cholesterol is several-fold less. This would also be consistent with the authors' observations that detect an increase in cellular TG and not cholesterol after incubating macrophages with VLDL (Fig 2c-d; Fig 3c-d). Mechanistically, this would mean that the function of LPL is compromised by VLDLR deficiency. In fact, VLDLR-deficient mice show extreme hypertriglyceridemia following an intragastric olive TG load (Goudriaan, *J Lipid Res* 2004). LPL is likely responsible for macrophage activation given that Angptl4 deficiency, which largely increases LPL levels and activity, causes extreme macrophage activation (Lichtenstein, *Cell Metab* 2010). To gain further insight into the mechanism underlying the effects of VLDLR deficiency, the authors should evaluate the effect of VLDLR overexpression and deficiency on LPL expression and activity, and evaluate the extent of selectivity of uptake of VLDL-TG versus VLDL-cholesterol (e.g. by using double-labeled human VLDL particles).
3. Substrate specificity of the VLDLR. The VLDR is involved in the clearance of TG derived from both VLDL and chylomicrons (Goudriaan, *J Lipid Res* 2004). The authors do not comment on a potential involvement of chylomicrons in VLDLR-dependent macrophage activation. Can similar effects be demonstrated on intracellular ceramides, and their involvement in expression of inflammatory pathways (Fig 6)? This would be important, as a high-fat diet as used in the *in vivo* experiments will result in high absorption of fat by the intestine that is transported towards the circulation by chylomicrons.
4. Translation of findings *in vitro* to the mouse. The authors convincingly show that the VLDLR is involved in the uptake of VLDL-derived lipids by macrophages *in vitro*, which causes production of inflammatory mediators, corroborating earlier findings (ref 28). A conceptually novel finding is that the VLDLR is largely involved in the accumulation of ceramides following uptake of VLDL-derived lipids via *de novo* synthesis. These ceramides may be intermediate molecules causing the

production of inflammatory mediators, as myriocin treatment partly reduces the induction of these inflammatory mediators (Fig. 6a). In an attempt to translate these findings to an in vivo setting, the authors perform a bone marrow transplantation to induce VLDLR deficiency in hematopoietic cells including macrophages. They demonstrate that hematopoietic VLDLR-deficiency in mice fed a HFD improves whole body glucose and insulin sensitivity (Fig 7). However, it is unclear whether similar mechanisms play a role as have been identified in vitro. Can the authors confirm similar effects on the cellular ceramide content in macrophages within white adipose tissue (EAT) as has been shown in vitro to confirm that reduced accumulation of ceramides in macrophages will improve insulin sensitivity in EAT (as speculated in lines 440-443)? Are similar effects observed in other white adipose tissue pads? Since bone marrow transplantation also results in full replacement of macrophages in the liver, improved insulin sensitivity of the liver may also be (partly) responsible for the beneficial metabolic effects. Can the authors provide data on macrophage concentration and activation on the liver following BMT? Ideally, a dual-isotope hyperinsulinemic euglycemic clamp analysis may provide information on effects of hematopoietic VLDLR-deficiency on hepatic versus peripheral insulin sensitivity.

Minor comments:

1. Source and concentration of VLDL. All in vitro experiments have been performed with a fixed concentration of 30 µg/ml human VLDL. Is this concentration based on protein or TG, and how does the concentration compare to the plasma concentration of VLDL? Was VLDL commercially obtained or isolated? How was oxidation excluded? Are any of the effects shown in vitro dose-dependent?
2. Expression analyses. It is unclear whether expression of the reported genes is sufficiently high to be relevant. Can the authors provide Ct values?
3. Line 101: Lipoproteins multi-molecular globular structures; not 'spheroid macromolecules'.
4. Line 103: Involvement of VLDLR in clearance of chylomicrons should in addition to VLDL should be acknowledged.
5. Line 131: If authors indeed used 'littermates', it should be specifically states that all mice were derived from a heterozygous breeding. This is unclear from the present description.
6. Line 144: Were peritoneal macrophages isolated after thioglycollate injection?

Reviewers' comments:

Reviewer #1 (Remarks to the Author):

In this paper, the Authors show that M1-like macrophages in obese adipose tissue express higher levels of VLDLR than M2-like macrophages. They then go on to use various in vitro and in vivo studies to show that the VLDLR is necessary for VLDL-mediated proinflammatory responses in macrophages. They also use chimeric mouse technology to demonstrate an effect of VLDL receptor macrophage deficiency on glucose tolerance and insulin sensitivity. As such, these studies build on previous papers showing that VLDLR KO protects mice from diet-induced obesity and that VLDL treatment promotes macrophage inflammation. The mechanistic aspects of this study relate to MAP kinase signaling and ceramide 16:0. The mechanistic studies are interesting but a bit thin.

Specific questions.

1. What is VLDLR expression in other macrophages from obese mice such as IP Macs and liver macrophages?

Answer 1: According to the reviewer's comment, we have examined the mRNA levels of VLDLR in other macrophages such as peritoneal macrophages and liver macrophages (kupffer cells). Peritoneal macrophages were obtained by thioglycolate injection, and kupffer cells were isolated from normal chow diet (NCD) or high fat diet (HFD) fed mice, according to previous report¹. As shown in Supplementary Fig. 2, the mRNA levels of VLDLR were elevated in both types of macrophages upon HFD. We include this information in the revised manuscript (p. 10, line 231-233).

2. While of interest, the Authors do not show whether MAP kinase activation is a result of ceramide synthesis, or a cause. These are straightforward studies and should be performed.

Answer 2: In order to investigate whether MAPK pathways may contribute to increase intracellular ceramide contents in the presence of VLDL, we measured the levels of cellular ceramides in VLDL-treated macrophages with or without MAPK inhibitors (SB203580, an inhibitor of p38 or SP600125, an inhibitor of JNK). As indicated in Supplementary Fig. 10, the levels of cellular ceramides remained unaltered with MAPK inhibitors in VLDL-treated macrophages. These data imply that MAPK inactivation, at least under this experimental condition, might not be sufficient to affect intracellular ceramide amounts in VLDL-treated macrophages. We provide this information in the revised manuscript (p. 21, line 481-485).

3. With respect to insulin resistance, do the Authors think that the ceramide synthesized within macrophages upon VLDL treatment can be released and taken up into insulin target cells causing insulin resistance?

Answer 3: To answer this comment, we examined the levels of released ceramides in conditioned media (CM). As shown in Supplementary Fig. 6, the levels of released ceramides were not significantly different in CM from WT and VLDLR KO bone marrow derived macrophages (BMDMs). These results propose that intracellular ceramides could potentiate inflammatory pathways in macrophages via VLDL-VLDLR axis. These new data are included in the revised manuscript (p. 14, line 322-325).

4. What is the effect of VLDL treatment on M2 macrophages with respect to inflammation and ceramide pathways compared to the M1 macrophages, since the M2 macrophages still express about 50% of the VLDLR?

Answer 4: To study whether VLDL might influence inflammation and ceramide synthesis in M2-like macrophages, we examined that the levels of cellular ceramides in M1- and M2-derived macrophages. As shown in Supplementary Fig. 5, VLDL did not affect inflammatory responses in M2-derived BMDMs. In addition, M2-derived BMDMs had less ceramide contents than M1-derived BMDMs. These findings are described in the revised manuscript (p. 14, line 309-311 and p. 19, line 448-449)

5. How do the Authors explain the decrease in adipose tissue macrophage numbers in the VLDLR KO chimeric mice? Something must be affecting chemotaxis, retention, or proliferation of macrophages.

Answer 5: We appreciate this comment. We also believe that this is an important issue. To investigate how

hematopoietic VLDLR deficiency could affect ATM numbers *in vivo*, BMT experiments are needed to be performed. However, it takes more than six months to perform the BMT experiment followed by the HFD challenge. Therefore, we have developed an alternative approach to address raised questions (Supplementary Fig. 8a). To delete the macrophages of recipient mice, WT mice were treated with clodronate. Donor bone marrow cells isolated from WT and VLDLR KO mice were pre-stained with one of the cell staining dyes, CellTracker™ (Thermo Scientific). Pre-stained donor bone marrow cells were adoptively transferred into recipient WT mice after 6 days of clodronate treatment. As shown in Supplementary Fig. 8b, the injected donor bone marrow cells were detected in recipient mice, and the degree of transferred bone marrow cells from either WT or VLDLR KO mice were not different. To determine whether hematopoietic VLDLR deficiency could affect infiltration or retention, blood monocytes obtained from GFP transgenic mice (GFP^{tg}) were injected into recipient mice 2 days after transferring bone marrow cells. Intriguingly, the number of GFP^{tg} monocytes in EATs of KO BMT was lower than that of WT BMT (Supplementary Fig. 8c). Recently, it has been suggested that macrophages are accumulated by retention as well as infiltration in obese adipose tissue^{2,3}. Conceptually, infiltrated macrophages in inflamed tissue flow into lymphatic vessel for recirculation in blood vessel⁴. To test the degree of macrophage retention, we investigated the several gene expression in mesenteric lymph nodes. Under current experimental scheme, we found that the mRNA levels of macrophage markers and GFP genes were not different in mesenteric lymph nodes from WT and KO BMT mice (Supplementary Fig. 8d). Thus, these data imply that macrophage retention might not be a key factor to change of ATM numbers upon hematopoietic VLDLR deficiency. Also, we investigated whether hematopoietic VLDLR deficiency might affect proliferation of ATMs. In EATs of WT and KO BMT mice, ATMs were stained with Ki67, as one of the cell proliferation marker genes. As shown in Supplementary Fig. 8e, the degree of Ki67 staining was not different in ATMs from WT and KO BMT mice. Together, these data imply that VLDLR deficiency would primarily influence macrophage infiltration into adipose tissue rather than retention or proliferation. These new findings are provided in the revised manuscript (p. 18, line 402-409).

6. In Figure 3G, it would be preferable to measure AKT phosphorylation as an indicator of insulin action.

Answer 6: According to this comment, we provide new western blot images and their quantitation data in a new Figure 3g (p. 12, line 282-283 and p. 13, line 284).

7. In Figure 3F, do the Authors have any ideas or data to show what factors are causing the changes in glucose transport. The introduction and discussion on this point are pretty generic and a bit out of date and do not take into account new data on macrophage-released factors. As the Authors undoubtedly know, there is a substantial literature on M2 macrophages and beige/BAT, possibly mediated through NE release. How does this play into the Authors studies?

Answer 7: According to the reviewer's suggestion, we analyzed the mRNA level of glucose transporter 4 (GLUT4). As shown in Supplementary Fig. 4, the level of GLUT4 mRNA was higher in adipocytes treated with conditioned media from VLDLR KO macrophages than WT macrophages. We include this information in the revised manuscript (p. 12, line 280-282).

Also, we carefully revised manuscripts including recent findings for M2 macrophages and their potential roles in beige/brown adipocytes activation (p. 3, line 80-84).

8. There might be a mistake in Figure 6, since panels A, (iNOS and TNF α) look to be identical.

Answer 8: We really appreciate this comment. During data processing, we made a mistake. We carefully corrected Figure 6.

9. In many places the Authors state that IL1 β causes insulin resistance. What is the evidence for this, since much data is contradictory on this point?

Answer 9: We also recognize that this is an important issue. In this study, we have considered IL-1 β as one of the pro-inflammatory "marker" genes. It has been demonstrated that various pro-inflammatory cytokines, including IL-1 β , have been implicated in disrupting insulin signaling^{5,6}. In addition, it has been reported that the blockade of IL-1 β signaling could reduce in systemic inflammation, eventually leading to improve insulin resistance^{7,8}. On the contrary, it has been reported that circulating levels of IL-1 β are not associated with the risk of type 2 diabetes⁹.

¹². Despite of these controversial issues on insulin sensitivity, we examined the mRNA level of IL-1 β as one of the markers for pro-inflammatory responses in this study.

10. The Authors should consider their results in light of the recent paper from the Glass laboratory on macrophage lipidomics (Oisi, et. al. *Cell Metab.* 25:42-427).

Answer 10: According to this comment, we include this reference in the revised manuscript (p. 3, line 95-97, and p. 4, line 98-101)

11. When the Authors measure macrophage TG content, does their method allow them to discriminate between fatty acids conjugated to glycerol vs. free intracellular fatty acids? Many of the effects the Authors describe could be due to increased intracellular SFA levels. This is an important distinction to try and make. It might be that the VLDLR is simply an added process to allow fatty acids to accumulate in macrophages.

Answer 11: In this study, we have utilized commercial kit (INFINITYTM, Thermo Scientific) to quantify the levels of triglycerides. The principle of this kit is to measure the released glycerol from triglycerides by lipase activation. Therefore, currently measured triglyceride levels in VLDL-treated macrophages cannot be regarded as the levels of intracellular fatty acids.

12. Is there any data on VLDL receptor in human obese fat? either from the Authors or in the literature?

Answer 12: According to the reviewer`s comment, we have examined the mRNA level of VLDLR in human adipose tissue. As shown in Supplementary Fig 1, the level of VLDLR mRNA in human adipose tissue showed a positive correlation with individual body mass index (BMI). We describe this information in the revised manuscript (p. 10, line 221-223).

13. In Figure 2E, have the Authors measured other inflammatory factors besides the four listed.

Answer 13: According to this comment, we measured the mRNA levels of IL-1 β and IFN γ genes. Additional data are included in the revised manuscripts (p. 11, line 246).

14. In their summary Figure 9, MAP kinase is not listed. Where do the Authors think this fits into this pathway?

Answer 14: According to this suggestion, we modified the summary Figure 9.

Reviewer #2 (Remarks to the Author):

Kyung Cheul Shin and colleagues investigated the role of the VLDL receptor (VLDLR) in lipid uptake, inflammation and M1/M2 polarisation of mouse macrophages *in vitro*, and determined the effect of hematopoietic VLDLR-deficiency on adiposity, glucose tolerance and insulin resistance in mice *in vivo*. They show that isolated macrophages take up VLDL-derived lipids which is coupled to ceramide production, expression of cytokines (e.g. iNOS, TNF α , and MCP1) and markers of M1 polarization. Most of these effects were abrogated or largely inhibited by VLDLR-deficiency. Hematopoietic deficiency for VLDLR did not affect high fat diet (HFD)-induced weight gain or fat mass, but improves fasting glucose and insulin and improved tolerance to glucose and insulin. The authors nicely demonstrate that part of the previously reported beneficial metabolic effects of VLDLR-deficiency (refs 27 and 28) can thus be conferred by hematopoietic cells, which may well be macrophages. The experiments seem to be adequately performed and in general the manuscript reads well. If strengthened, the data will be of interest to a wide public. I do have some reservations and comments as outlined below.

Major comments:

1. Novelty of the findings. Nguyen et al (ref 28) previously showed that VLDLR-deficient macrophages, upon incubation with VLDL, accumulate less lipids, have lower TNF α and IL-6 expression and produce less IL-6 and MCP-1 protein *in vitro*. Also, Goudriaan et al (ref 27) has demonstrated that whole-body VLDLR-deficiency markedly improves glucose tolerance in mice. The present data are thus not unexpected, which may compromise novelty to some extent.

Answer 1: In this study, we have explored the roles of macrophage VLDLR in the process of obesity-induced insulin resistance. In the absence of metabolic stress such as HFD, VLDLR whole body knockout (VLDLR KO) mice exhibit a similar extent of glucose tolerance compared with WT mice¹³. In DIO, however, VLDLR KO mice are glucose tolerant and sensitive to insulin action in peripheral tissues^{13,14}. It has been recently reported that chronic low-grade inflammation in adipose tissue is attenuated in HFD-fed VLDLR KO mice¹⁴. However, it remains largely unknown whether VLDLR-mediated VLDL uptake in macrophages would be a pivotal factor in contribution to obesity-induced insulin resistance via adipose tissue inflammation. Here, several lines of evidence suggest that macrophage VLDLR might play crucial roles in the progress of adipose tissue inflammation in obesity. First, we observed that the expression level of VLDLR in ATMs was increased in obese animals, particularly in M1-like macrophages. Second, macrophage VLDLR augmented M1-like macrophage polarization by uptaking VLDL. Third, hematopoietic VLDLR deficiency relieved adipose tissue inflammation and improved insulin resistance in DIO. Finally, we found that macrophage VLDL-VLDLR axis would be an important pathway to regulate cellular ceramides to mediate inflammatory responses. Therefore, we believe that this study clearly adds further understanding by revealing that macrophage VLDLR would be one of the key players to confer chronic inflammation and insulin resistance in obesity.

2. Mode of action of VLDLR in VLDL uptake. The authors quite consistently mention that the VLDLR is involved in the 'uptake of VLDL'. This is only one of two general views. Another view is that the VLDLR functions as a docking protein, ensuring binding of triglyceride (TG)-rich lipoproteins (TRL) to the cell surface in vicinity of lipoprotein lipase (LPL; the expression pattern of both proteins is similar), allowing LPL to selectively delipidate TRL. The latter mechanism may indeed explain that TRL selectively donate TG-derived fatty acids to metabolically active tissues, while the uptake of TRL-derived cholesterol is several-fold less. This would also be consistent with the authors' observations that detect an increase in cellular TG and not cholesterol after incubating macrophages with VLDL (Fig 2c-d; Fig 3c-d). Mechanistically, this would mean that the function of LPL is compromised by VLDLR deficiency. In fact, VLDLR-deficient mice show extreme hypertriglyceridemia following an intragastric olive TG load (Goudriaan, *J Lipid Res* 2004). LPL is likely responsible for macrophage activation given that Angptl4 deficiency, which largely increases LPL levels and activity, causes extreme macrophage activation (Lichtenstein, *Cell Metab* 2010). To gain further insight into the mechanism underlying the effects of VLDLR deficiency, the authors should evaluate the effect of VLDLR overexpression and deficiency on LPL expression and activity, and evaluate the extent of selectivity of uptake of VLDL-TG versus VLDL-cholesterol (e.g. by using double-labeled human VLDL particles).

Answer 2: We also believe that this is an important issue how macrophage VLDLR might process VLDL. According to this reviewer's comment, we studied the levels of lipoprotein lipase (LPL) mRNA expression and

its enzymatic activity in WT and VLDLR KO mice. As shown in Supplementary Fig. 3, LPL mRNA level and its enzymatic activity were not different in macrophages from WT and VLDLR KO mice. Furthermore, we found out that suppression of LPL expression via siRNA did not significantly influence VLDL uptake in macrophages. Although it has been reported that the enzymatic activity of LPL in serum is suppressed in VLDLR whole body knockout mice^{15,16}, it is unclear which cell type might be attributable to reduce LPL activity. Here, our data suggest that LPL may not be an essential factor to mediate VLDL uptake through VLDLR, at least, in macrophages. We include these data in the revised manuscript (p. 12, line 260-266)

3. Substrate specificity of the VLDLR. The VLDLR is involved in the clearance of TG derived from both VLDL and chylomicrons (Goudriaan, J Lipid Res 2004). The authors do not comment on a potential involvement of chylomicrons in VLDLR-dependent macrophage activation. Can similar effects be demonstrated on intracellular ceramides, and their involvement in expression of inflammatory pathways (Fig 6)? This would be important, as a high-fat diet as used in the *in vivo* experiments will result in high absorption of fat by the intestine that is transported towards the circulation by chylomicrons.

Answer 3: We appreciate this comment. To tackle this critique, WT and VLDLR KO macrophages were treated with chylomicron and subjected to determine the levels of inflammatory cytokine gene expression and intracellular ceramides. Unlike VLDL, the expression of pro-inflammatory cytokine genes was not greatly modulated by chylomicron in WT and VLDLR KO macrophages (Supplementary Fig. 7). In addition, there was no significant difference in ceramide contents between WT and VLDLR KO macrophages. These data are included in the revised manuscript (p. 15, line 348-353)

4. Translation of findings *in vitro* to the mouse. The authors convincingly show that the VLDLR is involved in the uptake of VLDL-derived lipids by macrophages *in vitro*, which causes production of inflammatory mediators, corroborating earlier findings (ref 28). A conceptually novel finding is that the VLDLR is largely involved in the accumulation of ceramides following uptake of VLDL-derived lipids via *de novo* synthesis. These ceramides may be intermediate molecules causing the production of inflammatory mediators, as myriocin treatment partly reduces the induction of these inflammatory mediators (Fig. 6a). In an attempt to translate these findings to an *in vivo* setting, the authors perform a bone marrow transplantation to induce VLDLR deficiency in hematopoietic cells including macrophages. They demonstrate that hematopoietic VLDLR-deficiency in mice fed a HFD improves whole body glucose and insulin sensitivity (Fig 7). However, it is unclear whether similar mechanisms play a role as have been identified *in vitro*. Can the authors confirm similar effects on the cellular ceramide content in macrophages within white adipose tissue (EAT) as has been shown *in vitro* to confirm that reduced accumulation of ceramides in macrophages will improve insulin sensitivity in EAT (as speculated in lines 440-443)? Are similar effects observed in other white adipose tissue pads? Since bone marrow transplantation also results in full replacement of macrophages in the liver, improved insulin sensitivity of the liver may also be (partly) responsible for the beneficial metabolic effects. Can the authors provide data on macrophage concentration and activation on the liver following BMT? Ideally, a dual-isotope hyperinsulinemic euglycemic clamp analysis may provide information on effects of hematopoietic VLDLR-deficiency on hepatic versus peripheral insulin sensitivity.

Answer 4: According to the reviewer's suggestions, total intracellular ceramide contents were determined in macrophages of EAT from BMT mice samples. In accordance with our *in vitro* experiments (Fig. 5), we observed that the levels of ATM ceramides from HFD-fed VLDLR KO BMT mice were decreased compared to those of HFD-fed WT BMT mice (Supplementary Fig. 11). These results are included in the revised manuscript (p. 22, line 449-500).

Regarding the issue of other fat pads, we examined the expression levels of pro-inflammatory cytokine genes in subcutaneous adipose tissue (SAT). Similar to EAT, SAT of KO BMT mice showed a tendency of reduce adipose tissue inflammation compared to that of WT mice in DIO (Supplementary Fig. 9). These results are included in the revised manuscript (p. 18, line 410-411).

As liver is another crucial organ to influence whole body insulin sensitivity, we examined the mRNA levels of macrophage and inflammatory marker genes in liver from BMT mice samples. Unlike adipose tissues, there was no significant difference in the levels of macrophage and inflammatory markers mRNA in liver from BMT mice samples. (Supplementary Fig. 9). This is probably due to the extremely low expression level of VLDLR in liver macrophage (kupffer) cells (Supplementary Fig. 2). These results are included in the revised manuscript (p. 18, line 411-413).

Unfortunately, we cannot provide the data using a dual-isotope hyperinsulinemic euglycemic clamp analysis due to lack of facility and experience. Instead, we examined the mRNA levels of gluconeogenic genes such as G6pase and PEPCK as they are closely associated with hepatic insulin resistance^{17,18}. As shown in Reviewer's Only Figure 1, there is no significant difference in hepatic gluconeogenic gene expression between WT and VLDLR KO BMT mice. These results suggest that adipose tissue inflammation would be a primary factor for obesity-induced insulin resistance in WT BMT mice compared to VLDLR KO BMT mice.

Reviewer's Only Figure 1. Hepatic gluconeogenic gene expression is not different between HFD-fed WT and KO BMT. Relative mRNA levels of gluconeogenic genes in livers of HFD-fed WT and KO BMT mice. Each mRNA level was normalized to cyclophilin mRNA. Data represent the mean \pm SD.

Minor comments:

1. Source and concentration of VLDL. All in vitro experiments have been performed with a fixed concentration of 30 μ g/ml human VLDL. Is this concentration based on protein or TG, and how does the concentration compare to the plasma concentration of VLDL? Was VLDL commercially obtained or isolated? How was oxidation excluded? Are any of the effects shown in vitro dose-dependent?

Answer 1: Human VLDL was purchased from Kalen Biomedical, which is described in "Methods". In this study, human VLDL (30 μ g/ml) was treated in macrophages. In human, VLDL is present at the range of 20 to 300 μ g/ml in plasma^{19,20}. Thus, we believe that 30 μ g/ml VLDL might be within physiological ranges. According to the manufacture's information, purchased human VLDL is a non-oxidized native form.

To determine VLDL dose, we tested various doses of VLDL and selected the concentration of 30 μ g/ml. We chose this concentration because the expression levels of pro-inflammatory genes began to change at this dose.

2. Expression analyses. It is unclear whether expression of the reported genes is sufficiently high to be relevant. Can the authors provide Ct values?

Answer 2: According to the reviewer's comment, we provided the Ct values in all the qPCR data.

3. Line 101: Lipoproteins multi-molecular globular structures; not 'spheroid macromolecules'.

Answer 3: Thanks for this critique. We amended the text to accommodate this. (p. 4, line 105).

4. Line 103: Involvement of VLDLR in clearance of chylomicrons should in addition to VLDL should be acknowledged.

Answer 4: Thanks for this comment. We carefully revised the manuscript (p. 4, line 107 and p. 19, line 428-430).

5. Line 131: If authors indeed used 'littermates', it should be specifically states that all mice were derived from a heterozygous breeding. This is unclear from the present description.

Answer 5: In this study, VLDLR-heterozygous mice were bred to generate WT and VLDLR-deficient littermates. To clarify this issue, we provide this information in the revised manuscript (p. 6, line 137-138).

6. Line 144: Were peritoneal macrophages isolated after thioglycollate injection?

Answer 6: Yes. As described in "Methods", peritoneal macrophages were isolated after thioglycollate treatment. This information is included in the revised manuscript (p. 7, line 159-160).

Reference

- 1 Bale, S. S., Geerts, S., Jindal, R. & Yarmush, M. L. Isolation and co-culture of rat parenchymal and non-parenchymal liver cells to evaluate cellular interactions and response. *Sci Rep* **6**, 25329, doi:10.1038/srep25329 (2016).
- 2 Ramkhelawon, B. *et al.* Netrin-1 promotes adipose tissue macrophage retention and insulin resistance in obesity. *Nat Med* **20**, 377-384, doi:10.1038/nm.3467 (2014).
- 3 Chung, K. J. *et al.* A self-sustained loop of inflammation-driven inhibition of beige adipogenesis in obesity. *Nat Immunol* **18**, 654-664, doi:10.1038/ni.3728 (2017).
- 4 Gordon, S. & Taylor, P. R. Monocyte and macrophage heterogeneity. *Nat Rev Immunol* **5**, 953-964, doi:10.1038/nri1733 (2005).
- 5 Jager, J., Gremeaux, T., Cormont, M., Le Marchand-Brustel, Y. & Tanti, J. F. Interleukin-1beta-induced insulin resistance in adipocytes through down-regulation of insulin receptor substrate-1 expression. *Endocrinology* **148**, 241-251, doi:10.1210/en.2006-0692 (2007).
- 6 Vandanmagsar, B. *et al.* The NLRP3 inflammasome instigates obesity-induced inflammation and insulin resistance. *Nat Med* **17**, 179-188, doi:10.1038/nm.2279 (2011).
- 7 Osborn, O. *et al.* Treatment with an Interleukin 1 beta antibody improves glycemic control in diet-induced obesity. *Cytokine* **44**, 141-148, doi:10.1016/j.cyto.2008.07.004 (2008).
- 8 Dinarello, C. A., Simon, A. & van der Meer, J. W. Treating inflammation by blocking interleukin-1 in a broad spectrum of diseases. *Nat Rev Drug Discov* **11**, 633-652, doi:10.1038/nrd3800 (2012).
- 9 Spranger, J. *et al.* Inflammatory cytokines and the risk to develop type 2 diabetes: results of the prospective population-based European Prospective Investigation into Cancer and Nutrition (EPIC)-Potsdam Study. *Diabetes* **52**, 812-817 (2003).
- 10 Interleukin 1 Genetics, C. Cardiometabolic effects of genetic upregulation of the interleukin 1 receptor antagonist: a Mendelian randomisation analysis. *Lancet Diabetes Endocrinol* **3**, 243-253, doi:10.1016/S2213-8587(15)00034-0 (2015).
- 11 Hajmrle, C. *et al.* Interleukin-1 signaling contributes to acute islet compensation. *JCI Insight* **1**, e86055, doi:10.1172/jci.insight.86055 (2016).
- 12 Maedler, K. *et al.* Low concentration of interleukin-1beta induces FLICE-inhibitory protein-mediated beta-cell proliferation in human pancreatic islets. *Diabetes* **55**, 2713-2722, doi:10.2337/db05-1430 (2006).
- 13 Goudriaan, J. R. *et al.* Protection from obesity in mice lacking the VLDL receptor. *Arterioscler Thromb Vasc Biol* **21**, 1488-1493 (2001).
- 14 Nguyen, A., Tao, H., Mettrione, M. & Hajri, T. Very low density lipoprotein receptor (VLDLR) expression is a determinant factor in adipose tissue inflammation and adipocyte-macrophage interaction. *J Biol Chem* **289**, 1688-1703, doi:10.1074/jbc.M113.515320 (2014).
- 15 Yagyu, H. *et al.* Very low density lipoprotein (VLDL) receptor-deficient mice have reduced lipoprotein lipase activity. Possible causes of hypertriglyceridemia and reduced body mass with VLDL receptor deficiency. *J Biol Chem* **277**, 10037-10043, doi:10.1074/jbc.M109966200 (2002).
- 16 Goudriaan, J. R. *et al.* The VLDL receptor plays a major role in chylomicron metabolism by enhancing LPL-mediated triglyceride hydrolysis. *J Lipid Res* **45**, 1475-1481, doi:10.1194/jlr.M400009-JLR200 (2004).
- 17 Accili, D. & Arden, K. C. FoxOs at the crossroads of cellular metabolism, differentiation, and transformation. *Cell* **117**, 421-426 (2004).
- 18 Jang, H., Lee, G. & Kim, J. B. SREBP1c-CRY1 axis suppresses hepatic gluconeogenesis upon insulin. *Cell Cycle* **16**, 139-140, doi:10.1080/15384101.2016.1235848 (2017).

- 19 Sattar, N. *et al.* Lipoprotein subfraction changes in normal pregnancy: threshold effect of plasma triglyceride on appearance of small, dense low density lipoprotein. *J Clin Endocrinol Metab* **82**, 2483–2491, doi:10.1210/jcem.82.8.4126 (1997).
- 20 National Cholesterol Education Program Expert Panel on Detection, E. & Treatment of High Blood Cholesterol in, A. Third Report of the National Cholesterol Education Program (NCEP) Expert Panel on Detection, Evaluation, and Treatment of High Blood Cholesterol in Adults (Adult Treatment Panel III) final report. *Circulation* **106**, 3143–3421 (2002).

REVIEWERS' COMMENTS:

Reviewer #2 (Remarks to the Author):

The authors have satisfactorily addressed most of my previous concerns. Apparently, the VLDLR on macrophages takes up lipoproteins as whole particles which is in contrast with the facilitating role of VLDLR in the selective delivery of triglyceride-derived fatty acids into adipocytes. It is reassuring to see that the effect of VLDLR-deficiency on ceramide accumulation in macrophages in vitro can be confirmed in vivo. I do have some reservations and comments as outlined below.

1. Line 350: The authors have now indicated the source of VLDL (Kalen Biomedical), but forgot to reveal the source of (human?) chylomicrons. A quick search learns that Kalen Biomedical does not provide chylomicrons, so how were chylomicrons obtained? If isolated from human blood, how were they purified from VLDL? Also, it is still unclear whether the concentrations used for VLDL and chylomicrons (30 µg/ml) refer to protein or triglycerides. Please specify in the manuscript.
2. Line 232. Apparently, VLDLR expression in liver macrophages was negligible. The authors may wish to specifically state this in the text.

Response to the Reviewers' Comments

MS ID#: NCOMMS-17-03982A

REVIEWERS' COMMENTS:

Reviewer #1 (Remarks to the Author):

No further comments.

Reviewer #2 (Remarks to the Author):

The authors have satisfactorily addressed most of my previous concerns. Apparently, the VLDLR on macrophages takes up lipoproteins as whole particles which is in contrast with the facilitating role of VLDLR in the selective delivery of triglyceride-derived fatty acids into adipocytes. It is reassuring to see that the effect of VLDLR-deficiency on ceramide accumulation in macrophages in vitro can be confirmed in vivo. I do have some reservations and comments as outlined below.

1. Line 350: The authors have now indicated the source of VLDL (Kalen Biomedical), but forgot to reveal the source of (human?) chylomicrons. A quick search learns that Kalen Biomedical does not provide chylomicrons, so how were chylomicrons obtained? If isolated from human blood, how were they purified from VLDL? Also, it is still unclear whether the concentrations used for VLDL and chylomicrons (30 µg/ml) refer to protein or triglycerides. Please specify in the manuscript.

Answer 1: Human VLDL was purchased from Kalen Biomedical (#770100) and human chylomicron was purchased from BioVision (#7285-1000). According to the manufacturer's information, purchased VLDL contains minimum 1.1 mg/ml protein and purchased chylomicron is composed of 98 % lipids and 2 % protein. According to the reviewer's comment, we provided this information in the revised manuscript (p. 20, line 472-476).

2. Line 232. Apparently, VLDLR expression in liver macrophages was negligible. The authors may wish to specifically state this in the text.

Answer 2: According to this comment, we modified the revised manuscript (p. 6, line 146-148).